

**StageIV-IRC: A High-resolution Dataset of Extreme Orographic Quantitative**
**Precipitation Estimates (QPE) Constrained to Water Budget Closure for**
**Historical Floods in the Appalachian Mountains**
Mochi Liao[1] and Ana P. Barros[1]
1. Civil and Environmental Engineering, University of Illinois Urbana-
Champaign, Urbana, IL
**Corresponding Author:**
Dr. Ana Barros
E-mail: barros@illinois.edu
Phone: +1 217-333-8038





## Abstract

Quantitative Flood Estimation (QFE) in complex terrain remains a grand challenge in operational hydrology due to the lack of accurate high-resolution Quantitative Precipitation Estimates (QPE) for operational forecasting and for calibrating hydrologic models. Here, we present a high-resolution (i.e., 250m, 5-minute-hourly) QPE dataset for 215 extreme rainfall events occurred in 26 gauged mountainous basins in the Appalachian Mountains from 2008 to 2024. This dataset is developed by applying inverse rainfall corrections (IRC) derived from physically-based rainfall-runoff modeling (Liao and Barros, 2022 and 2023) to the Next Generation Weather Radar (NEXRAD) Stage IV analysis (4 km resolution, hourly). The corrected Stage IV analysis QPE is referred to as StageIV-IRC (StageIV with Inverse Rainfall Correction). The unique advantage of this StageIV-IRC QPE dataset is its agreement with ground-based rainfall measurements while achieving water budget closure at the storm-flood event scale within observational uncertainty of streamflow observations, which is the gold standard in hydrological modeling. This dataset is the first QPE dataset aiming to improve QFE in the complex terrain by reducing biases for extreme precipitation events, and it can be used to evaluate the skill of hydrologic models in the same basins and support model calibration. The StageIV-IRC QPE dataset is publicly available at https://doi.org/10.5281/zenodo.14028866, and improved initial soil moisture maps for the studied extreme precipitation events, derived from the same IRC framework, are available in the same repository (Liao and Barros, 2025c).



## 1. Introduction

Over the past few decades, extreme precipitation has become an increasingly important
research topic due to its social, economic, and environmental impacts (e.g., Alimonti et al., 2022;
Wernberg et al., 2013). Studies show that both total annual precipitation and extreme precipitation
events have increased in the US and in other parts of the world during the last century (e.g., Milly
et al., 2002), often resulting in floods (e.g., Pielke and Downtown, 2002), and flash floods in the
context of complex terrain due to steep slopes (e.g., Schumacher, 2017; Czigány et al., 2010).
Flash floods are characterized by fast rainfall-runoff responses on the scale of a few hours (< 6
hours) after extreme precipitation events for watershed areas often ranging from a few tens to
hundreds of square kilometers (e.g., Borga et al., 2014; Lumbroso and Gaume, 2012). As one of
the deadliest natural hazards, flash floods are often associated with landslide events (e.g., Tao and
Barros, 2014;  Gupta et al., 2016; Deijns et al., 2022) and cause loss of life and property damage
(Špitalar et al., 2014), such as recently in the last three years in the Appalachian Mountains, USA,
and in Southern Spain. Despite extensive studies to improve flash flood simulations in small
headwater basins, hydrological skill scores (e.g., Kling-Gupta Efficiency or KGE) remain poor at
event scales largely due to significant difficulties involved in estimating highly localized
orographic precipitation in complex terrain, which in turn implies that hydrologic models are not
calibrated using forcing representative of realistic extreme events (e.g., Andrieu et al. 1997;
Huffman et al., 2007; Mtibaa and Asano, 2022).
Current approaches involved in precipitation measurement and Quantitative Precipitation
Estimation (i.e., QPE) broadly include in-situ point-scale observations using rain gauges and
disdrometers, and remote spatial observations using ground-based radar and space-based sensors.
In complex terrain, there is often a scarcity of in situ measurements due to difficult access. For





example, the rain gauge network from NASA's Integrated Precipitation and Hydrology Experiment

is the only relatively dense rain gauge network installed at high elevations in the entire

Appalachians (e.g., Barros et al. 2014). Other QPE products (e.g., radar QPE data) are plagued by

uncertainties from various sources (e.g., ground clutter artifacts, retrieval uncertainties, and radar

viewing geometry (Villarini and Krajewski, 2010; Arulraj and Barros, 2021; Kreklow et al., 2020;

Huffman et al., 2007; Andrieu et al., 1997; Durden et al., 1998). Numerical weather prediction

(NWP) is an alternative to measurement. However, QPE products from NWP models are

characterized by significant uncertainties when evaluated against rain gauges (e.g., Zhang and

Anagnostou, 2019), leading to large flood simulation errors when used as inputs to hydrological

models, or introducing large structural uncertainty when used for model calibration (e.g., Tao et

al., 2016; Weiland et al., 2015; Diomede et al., 2008; Kobold and Suselj, 2005). Due to these

uncertainties and errors involved, focus has been directed towards enhancing QPE using various

methods: data merging of raingauge and radar precipitation (e.g., McKee and Binns, 2016;

Goudenhoofdt and Delobbe, 2009; Delrieu et al., 2014; Nanding et al., 2015; Sideris et al. 2013;

Schiemann et al. 2011), combined radar reflectivity and retrieval corrections (e.g., Vignal et al.,

2000; Shao et al., 2021; Dinku et al., 2002), and data assimilation into NWP models (e.g.,

Rafieeinasab et al., 2015; Wehbe et al., 2020). Rain gauge and disdrometer measurements are often

used as references for these QPE optimization approaches (e.g., Harrison et al., 2000; Shao et al.,

2021; Fulton et al., 1998). The 'ground truth', however, has its own error (e.g., spatial

representativeness, wind artifacts around the gauge orifice, and calibration, among others;

Kochendorfer et al., 2017), and fails to capture highly localized orographic enhancement (e.g., Prat

and Barros, 2010b; Gentilucci et al., 2021; Buytaert et al., 2006). Gauge-radar fusion often relies

on geostatistical assumptions that are primarily distance-based (e.g., Areerachakul et al., 2022;



Cassiraga et al., 2021; Wang et al., 2020; Maggioni and Massari, 2018), lacking the full picture of
complex basin topography, which has a regulating role in orographic precipitation processes.

To address this long-standing QPE challenge in complex terrain, a general QPE error

quantification framework was developed leveraging widely available quality United States
Geological Survey (USGS) streamflow observations at the outlet of headwater basins in complex
terrain, consisting of 2 distinct paths: 1) rain gauge bias correction, and 2) grid-level QPE
correction constrained to watershed-scale water budget closure. The first pathway includes rain
gauge bias corrections at gauge locations both at the diurnal and climate scales, and the
geostatistical distribution of rain gauge biases across a basin. The second pathway includes an
innovative inverse QPE correction method by backward propagating runoff uncertainty using a
hydrological model via streamlines to precipitation at storm-event scale, and the methodology is
termed Inverse Rainfall Correction (IRC), which is developed by the same authors (Liao and
Barros, 2022 or LB22).

LB22 found that initial soil moisture uncertainty causes inferior performance of IRC

because large initial condition errors lead to significant uncertainties in travel time distributions.
Soil moisture is considered a particularly important factor among soil properties due to its
significant role in affecting the generation of runoff, hence dramatically altering the timing of flood
front and its magnitudes (e.g., Vivoni et al., 2007; Marchi et al., 2010; Penna et al., 2011), and soil
moisture can vary dramatically at hourly timescales, changing from fully saturation levels to
wilting point levels conditional on the specific texture and other properties of the soils (Grillakis
et al., 2016). Initial soil moisture conditions can therefore determine whether a rainstorm produces
a major flash flood or not (e.g., Komma et al., 2007; Zehe and Blöschl, 2004). However, due to
the limited availability of soil moisture sensors, there are not many studies quantifying the impact



of soil moisture on runoff simulation (e.g., Silvestro et al., 2019; Laiolo et al., 2016; Zappa et al.,
2011; Uber et al., 2018). Liao and Barros (2025b) developed an Initial Condition Correction (ICC),
which is based on travel time distributions and is coupled with the general IRC approach,
demonstrating large improvements in initial soil moisture estimation. Note that when
implementing the IRC and ICC, we are using a fully distributed physics-based uncalibrated model
(i.e. Duke Coupled Hydrological Model, DCHM) that has been used successfully for more than
two decades for hydrologic studies in the Southern and Central Appalachians (e.g., Tao and Barros,
2013, 2014, 2018 and 2019; Tao et al. 2016; Yildiz and Barros 2004, 2007 and 2009), and
consequently uncertainty from model structure and model parameters is assumed to be small.
Hydrological model parameters certainly have an impact on rainfall-runoff response, but they are
generally only of secondary importance compared to the precipitation proper and antecedent soil
moisture distributions, especially for smaller basins (e.g., Dobler et al., 2012; Mockler et al., 2016).
In this work, IRC and ICC are combined into one structure (referred to as the IRC-ICC
framework) to construct an improved QPE dataset aiming to close the water budget at the scale of
storm-flood events along the entire Appalachian Mountains (e.g., Liao and Barros 2022 and
2025b). The study region is set to be the Appalachian Mountains because they are prone to extreme
precipitation and flash floods due to orographic lift of moisture-laden air masses coming from the
Gulf of Mexico and the Atlantic Ocean (e.g., Troch et al., 1994; Smith et al., 2011; Liao and
Barros, 2023). A recent example is Hurricane Helene, which caused over 200 deaths and over $50
billion in property damage in the Southeast US in September 2024. The IRC-ICC framework is
employed in 26 headwater basins and 215 extreme events (during 2008-2024) using the Next
Generation Weather Radar (NEXRAD) StageIV dataset as original inputs, at a spatial and temporal



resolution of 250 m and 5 minutes, respectively, and the improved post IRC-ICC QPE data (i.e.,
StageIV-IRC) are made available in this study.
The manuscript is organized as follows. The data sources and the QPE error quantification
framework, which consists of rain gauge bias correction and the IRC-ICC framework, are detailed
in Section 2. Section 3 presents this new dataset (StageIV-IRC) along with data assessment from
various aspects. Section 4 discusses the potential application of this new dataset and future work.
Section 5 provides access to the dataset and a summary of the work.

## 2. Data and Methodology


**2.1 Radar QPE StageIV**


The NCEP/EMC StageIV is a precipitation estimation product, developed using hourly and
6-hourly radar-raingauge precipitation analyses at regional scales (Lin and Mitchell, 2005). In
complex terrain, it is known that radar QPE suffers from the blockage of topography, overshooting
and retrieval uncertainties, leading to large uncertainties in rainfall estimation. In 2007, as part of
the ground validation (GV) of the Precipitation Measurement Missions (PMM) program by NASA
(e.g., Prat and Barros, 2010a and 2010b), 34 tipping bucket raingauges were installed in the
Southern Appalachians and have been well-maintained since 2007 (e.g., Barros et al., 2014). In
this work, raingauge measurements from a GV raingauge network in the Southern Appalachians
are utilized to reduce StageIV uncertainties.

**2.2 GV Rain Gauge Observations**


A rain gauge network in support of PMM GV was installed in the Pigeon River basin for
the 10 year 2007-2018 period (Barros et al. 2014). A map of this rain gauge network is plotted in



Figure 1. Every rain gauge is labelled with a number, and exact locations are documented in Table
1. This rain gauge network is regularly visited and maintained at least three times a year, including
on-site cleaning and calibration. In this study, these rainfall measurements are used as a basis to
adjust hourly StageIV QPE. Note these rain gauge measurements can be downloaded at
http://dx.doi.org/10.5067/GPMGV/IPHEX/GAUGES/DATA301 (Barros et al., 2017). Besides
rain gauges, a network of Parsivel disdrometers was installed during 2013-2014, with each
disdrometer location denoted by the letter P in Figure 1. These disdrometer data were only used
for independent evaluation because of short records. It is worth noting that rain gauges are installed
mostly along the ridges while disdrometers are generally located at lower elevations.

<Figure 1 here please>

**2.3 Methodology**
The methodology of this work includes three major components: 1) rain gauge bias
correction, 2) grid-scale QPE correction by closing the water budget using stream gauge
measurements, and 3) basin and event selection procedures and model setup.
**2.3.1 Rain gauge Bias Correction**
A schematic drawing of the rain gauge bias correction framework to derive gauge-
improved QPE (named StageIV$_{DBKC}$) is provided in Figure 2.
<Figure 2 here please>





166  First, to make meaningful comparison between StageIV and rain gauges in space, a fractal

167 downscaling algorithm is used to create StageIV$_D$ at 1km from the original StageIV at 4 km

168 resolution. Subsequently, bias correction using raingauge measurements is employed to create

169 StageIV$_{DB}$ at hourly timescales. StageIV$_{DB}$ data are then evaluated against the rain gauge

170 climatology from 2008 to 2017 to reduce biases that depend on weather regime, and climatological

171 biases are then interpolated using the ordinary Kriging method. The resulting dataset is named

172 StageIV$_{DBKC}$ (abbreviated as STIV$_{DBKC}$).

**174 2.3.2 Fractal downscaling**

175  The methodology for fractal downscaling was first proposed by Bindlish and Barros (1996)

176 and subsequently demonstrated through various applications to precipitation downscaling from

177 models (Bindlish and Barros, 2000) and remote sensing data (Nogueira and Barros, 2015; Tao and

178 Barros, 2010). Here, a brief description is presented.

179  The assumption of self-similarity is imposed in fractal downscaling approach. The

180 parameters used in this approach involve: fractal dimension $D$, Hurst coefficient $H$, and the spectral

181 exponent β that are related through the following equations:

$$D = \frac{7-\beta}{2} \tag{1}$$

$$H = \frac{\beta-1}{2} \tag{2}$$

184  The parameter β describes rainfall statistics across different spatial scales, and it is

185 calculated as the slope of the power spectral density curve in the 2D Fourier domain of the rainfall

186 field (log-log plot). The parameter H is the Hurst coefficient which is a measure of autocorrelation



strength with higher value representing stronger autocorrelation. The power spectral density of a
2D field in Fourier domain is calculated as the following:
$$Z(u,v) = \left(\frac{L}{N}\right)^2 \sum_{x=0}^{N-1} \sum_{y=0}^{N-1} z(x,y)\, exp\left[-\frac{2\pi i}{N}(ux+vy)\right] \tag{3}$$

where N is the total number of grid points of the rainfall field z(x,y) with grid size being
the L, u and v correspond to frequency indices in the Fourier domain in each direction. The
averaged power spectral density is given:
$$S_j = \frac{1}{L^2 N_j} \sum_{1}^{N_j} |Z(u,v)|^2 \tag{4}$$

where $N_j$ denotes the number of points that meet the following condition: $j < \sqrt{u^2+v^2} <$
$j+1$. There is roughly a power-law relationship between the wavenumber k and the mean power
spectral density, and k is defined as below:
$$k = \frac{2\pi}{\sqrt{u^2+v^2}} \tag{5}$$

$$S \sim k^{-\beta-1} \tag{6}$$

Specifically, the corresponding $S$ value when wavenumber k = 1 is referred to as the
roughness factor, which is a representation of the variance of the field.
Assuming rainfall fields have self-similar statistics from coarse resolution to fine
resolution, then fine scale rainfall fields can be generated by preserving these self-similar statistics.
This is accomplished by creating a Brownian surface at desired fine scale resolution while sharing
the same spectral slope and roughness factor as the original rainfall field based on Bindlish and
Barros (1996):





$$Z_D(u,v) = \frac{Z_b(u,v)}{k_r^{(\beta-\beta_b)/2}} exp\left[\frac{1}{2}\left(S_{r,1} - \frac{\beta+1}{\beta_b+1}S_{r,2}\right)\right] \tag{7}$$

where $\beta$, $\beta_b$, $Z_D(u,v)$ and $Z_b(u,v)$ are the spectral slope of 2D original rainfall field, the
spectral slope of the Brownian surface, interpolation surface in the Fourier domain and original
Brownian surface, respectively; $k_r$ is the wavenumber and $S_{r,1}$ and $S_{r,2}$ are the roughness factors
of the 2D original rainfall fields and Brownian surface. Due to the non-uniqueness of Brownian
surfaces, multiple replicates of interpolation surfaces $Z_D$ can be obtained. In this study, an
ensemble of ND interpolation surfaces is derived, thus ND rainfall fields at finer resolution
preserving the same rainfall statistics at coarse resolution are generated. In this work, a total of
ND=50 ensemble downscaled rainfall fields are created similar to Nogueira and Barros (2015),
and a series of rainfall bias correction steps is described in Figure 2 and is applied to the ensemble
mean of the downscaled rainfall fields.

**2.3.3 Bias Correction**
The _first_ phase of bias correction is carried out at the event scale: a linear regression is
established between rain gauge measurements and collocated downscaled radar pixel estimates
using the following formula:
$$R_g^t(i_g, j_g) = \kappa R_r^t(i_g, j_g) + \varepsilon \tag{8}$$

where $R_r$ and $R_g$ represent radar and rain gauge measurements respectively, $\kappa$ and $\varepsilon$ are the
slope and the intercept of a polynomial fit between $R_r$ and $R_g$. Hourly StageIV$_D$ estimates and
corresponding rain gauge observations in the same StageIV$_D$ pixel were identified if at least 2 rain
gauges in the same StageIV$_D$ pixel measure non-zero rainfall. A linear regression was applied to



all StageIV$_D$ pixels within one standard deviation of the regression line at an hourly timescale by
assuming homogeneity of variances or homoscedasticity.
The *second* phase of bias correction is done at decadal scale: aiming to reduce systematic
radar errors caused by retrieval uncertainties and viewing geometry in complex terrain,
demonstrating strong diurnal (time of day) and seasonal (weather regime) error dependencies due
to miss detection of shallow rainfall systems related to radar overshooting in the Southern
Appalachian when comparing against 10-year rain gauge observations (e.g., Wilson and Barros,
2014; Arulraj and Barros, 2017). For this purpose, when rain gauge observations are <2mm/hr and
Stage IV$_D$ estimates are 0mm/hr, the StageIV$_D$ value was automatically replaced by the rain gauge
observations, which is referred to as the Light Rainfall Correction (LRC). Moreover, if StageIV$_D$
estimates equal to 0 where at least one collocated rain gauge observation is >2mm/hr, then
StageIV$_D$ estimates are replaced by the mean of all collocated rain gauge observations, namely
Mean Rainfall Correction (MRC). Lastly, for highly localized precipitation (i.e., less than 2
rainguages register nonzero rain in the study domain) which is normally produced by convective
activity, the rainfall differences between the StageIV$_D$ and the local rain gauge observations were
bilinearly distributed across nearby 25 grids (a 5x5 grid square centered at the StageIV$_D$ pixel)–
Convective Rainfall Correction (CRC). For most of the raining hours, there are more than 2 rain
gauges recorded rain, in which case the differences at each pixel between radar estimates and
raingauge measurements were spatially interpolated using a geostatistical interpolation method
(e.g., ordinary Kriging), which is refered to as the Global Rainfall Correction (GRC).





**2.3.4 Ordinary Kriging**
Ordinary Kriging is a geostatistical interpolation method that generates artificial values of
a variable at a specific location, aiming to minimize spatial variance. In this work, rainfall
differences between raingauge observations and StageIV$_{DB}$ are calculated and distributed across
the entire basin using a spatial variance model, which is commonly referred to as a semi-variogram
model. Specifically, a spherical semi-variogram model is used. Literature regarding the choice of
semi-variogram models and their properties can be found (e.g., Li and Heap, 2008; Oliver and
Webster, 2015; Zimmerman and Zimmerman, 1991). Bohling (2005) pointed out that spherical
models reach the maximum variance for relatively shorter spatial lags, therefore more suitable to
capture highly nonlinear and localized orographic precipitation (McBratney and Webster, 1986):
$$\gamma(h) = C_0 + (C - C_0)\left(\frac{3h}{2d} - \frac{1}{2}\left(\frac{h}{d}\right)^3\right) \text{ if } 0 \le h \le d \qquad (9.1)$$

$$= C \qquad\qquad\qquad \text{ if } h > d \qquad (9.2)$$

$$\gamma_{0i} = \frac{1}{N_A}\sum_{k=1}^{N_A}\gamma_{ki} \qquad (9.3)$$

$$\gamma_{00} = \frac{1}{N_A}\sum_{k=1}^{N_A}\sum_{l=1}^{N_A}\gamma_{kl} \qquad (9.4)$$

where h is the lag, d is the range, C and $C_0$ are the sill and nugget values of the semi-variogram
model, $N_A$ is the number of raingauges. The nugget is assumed to be zero if local variability and
measurement error are neglected at the point scales (Diggle and Ribeiro, 2007). The interpolated
rainfall difference at a location $x_0$ $Z_{ok}^*(x_0)$ is calculated using a weighted combination of all
available differences multiplied by Kriging weights:
$$Z_{ok}^*(x_0) = \sum_{i=1}^{n}\lambda_i^{ok}G(x_i) \qquad (10.1)$$



$$\sum_{i=1}^{n} \lambda_i^{ok} = 1 \qquad (10.2)$$

Optimal Kriging weights can be obtained by a series of linear equations using the Lagrange
multiplier $\mu$ method:
$$\begin{pmatrix} \gamma_{11} & \cdots & \gamma_{n1} & 1 \\ \vdots & \ddots & \vdots & \vdots \\ \gamma_{1n} & \cdots & \gamma_{nn} & 1 \\ 1 & \cdots & 1 & 0 \end{pmatrix} \begin{pmatrix} \lambda_1^{OK} \\ \vdots \\ \lambda_n^{OK} \\ \mu \end{pmatrix} = \begin{pmatrix} \gamma_{01} \\ \vdots \\ \gamma_{0n} \\ 1 \end{pmatrix} \qquad (11)$$

In this work, Ordinary Kriging interpolates differences between radar data and raingauge
observations to produce gauge-corrected STIV$_{DBKC}$ dataset.

**2.3.5 Precipitation Assessment Metrics**
Assessment metrics include the following: bias and root mean square error between radar
estimation and raingauge measurement, false alarm rate, the probability of detection, threat score
and Heidlke skill score, following McBride and Ebert, 2000. An instance when both radar QPE
and rain gauge observation exceed a specified rain rate threshold is a hit (H); when observation
matches the criterion and radar QPE does not, it is classified as a miss (M); if the opposite happens,
then it is a false alarm (FA). The calculation of these metrics relied on a collection of Hs, Ms, and
FAs:
$$Bias = \frac{1}{N}\sum_{n=1}^{N}(O_n - R_n) \qquad (12)$$

$$RMSE = \sqrt{\frac{1}{N}\sum_{n=1}^{N}(O_n - R_n)^2} \qquad (13)$$

$$FR = \frac{FA}{H+FA}, 0 \le FR \le 1 \qquad (14)$$





$$PD = \frac{H}{H+M}, 0 \leq PD \leq 1 \tag{15}$$


$$TS = \frac{H}{H+FA+M}, 0 \leq TS \leq 1 \tag{16}$$


$$HSS = 2 * \frac{Z*H-FA*M}{((H+FA)*(Z+FA))+((M+H)*(M+Z))}, -1 \leq HSS \leq 1 \tag{17}$$


where O is the rain gauge observation, R is the radar QPE, and N is the number of points. Z
represent the number of zeros, meaning both raingauge and radar do not register a rainfall record
above a predefined threshold. A threat score (TS) of 0.5 means over 50% of cases meet the
criterion, and the higher the better. An HSS of 0 means a forecast has the same performance as a
random guess.

**2.3.6 Inverse Hydrologic Correction**

At flash flood timescales in headwater basins, streamflow uncertainty and precipitation

uncertainty are strongly connected in a nonlinear way through rainfall runoff processes. Liao and
Barros (2022) developed a Lagrangian-based framework named Inverse Rainfall Correction (IRC),
allowing backpropagating streamflow uncertainty to precipitation inputs in space and time through
an uncalibrated distributed hydrological model (i.e., DCHM), achieving water budget closure at
the event scale in small headwater basins. As stated earlier, the uncertainties associated with
parameters and the hydrological model DCHM are neglected since the model configurations have
been used and improved over the past two decades for this region accounting for various soil,
vegetation, and river processes (e.g., Tao and Barros, 2013, 2014, 2018 and 2019; Yildiz and
Barros, 2005 and 2007; Lowman and Barros, 2016), and the IRC framework has been tested in




multiple headwater basins extensively in this region with consistent success. The detailed
description of the IRC is provided in Section 2.3.8 and Appendix A.

It is worth noting that IRC is a general framework to improve QPE at the watershed scale

that can be incorporated into any distributed hydrological models. Liao and Barros (2025a, 2025b)
investigated the impact of model structure uncertainty and initial condition uncertainty on IRC and
then the downstream product: the resulting IRC-improved QPE. The results suggest with improved
watershed physics at finer resolution (e.g., river bank storage, Liao and Barros, 2025a), river
routing algorithms (e.g., XY routing, Liao and Barros, 2025a) and improved antecedent soil
moisture distributions (Liao and Barros, 2025b), post-IRC QPE demonstrate realistic precipitation
features at high resolution that are aligned with basin topography with ridges associated with
higher precipitation than valleys in general, showing a significant improvement from the original
StageIV dataset which is characterized by unnatural boxy precipitation patterns in complex terrain
due to resolution issues and over or underestimation depending on topography and distance from
the radar site.

As briefly mentioned before, LB22 reviewed various sources of uncertainty that can

prevent post-IRC QPE from achieving water budget closure, among which initial condition
uncertainty in soil moisture is a noteworthy source. Improved initial condition estimation results
in significantly improved post-IRC precipitation features in complex terrain by better capturing
transient travel time distributions (Liao and Barros, 2025b). They found that the uncertainty tied
to initial conditions is more significant for less extreme events. Nevertheless, the initial condition
correction method is coupled with the IRC framework, and the complete framework is named the
IRC-ICC framework. The specifics regarding the IRC, ICC, and IRC-ICC are schematically drawn
in Figure 3.




\<Figure 3 here please\>

Using the definitions of characteristic timings shown in panels c) and d), characteristic flow

regime windows are identified. In principle, the number and the size of the windows depend on
the complexity of the hydrograph. ICC is only applied to windows 2 and 5 in this example, which
represents a segment of the hydrograph characterized by the differences between rising points in
observations and simulations, and a segment characterized by slow recession, respectively. The
assumption is that precipitation uncertainty regulates streamflow differences during peak flows
(i.e. windows 3 and 4). $W_{nm}$ represents the framework state after window m for iteration n. The
resolution settings for the DCHM are: spatial resolution: 250m, and temporal resolution: 5 minutes.
**2.3.7 Implementation of Lagrangian Tracking**

A flood event is simulated by the DCHM at the basin outlet with grid-based time-varying

velocity fields for different soil layers. When the precipitation starts (i.e. basin-averaged
precipitation > 0.1mm/hr), new particles (passive tracers) are launched at the same frequency of
model temporal resolution (5 minutes), but only at non-zero precipitation grids in all soil layers
following the velocity fields calculated by the DCHM, and the tracking resolution is 10 seconds,
amounting to a release of approximately 600,000 particles for basin with an area of 120km$^2$ over
a 24-hour period. During the tracking phase, each particle is saved along with information
regarding its source location (grid-point where it originates), time of release ti, and travel time tT
(tT is defined as the difference between current time t and the time of release ti, i.e., tT = t – ti ).
Multiple particles from different source locations can have the same travel time, which is the basis





for identifying the number of trajectories contributing to the hydrograph at the outlet as a function
of time.

### 2.3.8 QPE Correction Using IRC

At time t, the water difference *wd(t)* between the observed and simulated streamflow over
the time *Δt* between two consecutive discharge observations represents the fraction of runoff that
eventually leaves the basin as streamflow. Errors in precipitation forcing propagate to the runoff,
under the assumption of negligible model and parameter uncertainties, *wd(t)* can be entirely
attributed to precipitation error, which is the focus of this work.

$$wd(t) = [Q_{obs}(t) - Q_{simu}(t)] \times \Delta t \tag{18}$$

The subscripts *obs* and *simu* refer to observed and simulated discharge, respectively.
The strategy for the inverse rainfall correction (IRC) using hydrograph analysis is to follow the
trajectories available from the Lagrangian tracking backward from the basin outlet to the source
locations at time $t_i$ and apply a correction at the source locations proportional to the original QPE
magnitude to reduce *wd* at time *t*. Detailed formulas with a conceptual drawing can be found in
Appendix A. The embedded assumption is that larger QPE values have larger uncertainties. Note
that QPE corrections that happened earlier in time will have an impact on runoff simulation at
future times, and this is the reason why the IRC framework is a recursive framework. The detailed
rainfall correction steps can be found in (Liao and Barros, 2022).



### 2.3.9 Methods for Reducing Uncertainties from Other Sources

As briefly mentioned before, uncertainties from other sources (e.g., model physics, model numerical formulation, antecedent soil moisture conditions, etc.) impact travel time distributions and simulated streamflow to a higher or lesser degree depending on location, antecedent conditions, and storm system. Previous studies demonstrate that, for flood-producing events in small headwater basins, streamflow response is largely controlled by precipitation inputs (e.g., Iwasaki et al., 2020). In this section, we briefly describe the methods used to minimize the impacts from other sources to enhance water budget closure using the IRC approach.

As discussed in the Introduction DCHM has been used in the Appalachian Mountains at event-scale (e.g., Tao and Barros, 2013, 2014, 2018 and 2019; Tao et al. 2016) and at seasonal and interannual scales (Yildiz and Barros 2005, 2007 and 2009), and thus extensive analysis of parameter uncertainty and model structure uncertainty has been conducted previously. Recent improvements to the flood routing algorithm have resulted in significant improvements in flood peak timing in headwater basins to reconcile the hydraulics of flood wave propagation on steep slopes at the highest elevations with milder slopes at intermediate elevations in the valleys (Liao and Barros, 2025a). Their results also suggest meandering effects, riverbank storage, and initial soil moisture distributions can impact the early rising period of the hydrographs. Significant and consistent improvements are made when introducing an initial condition correction (ICC) module to reduce initial condition uncertainty (Liao and Barros, 2025b). This innovative ICC module is coupled with the IRC framework. The red arrows in Figure 3e indicate where ICC is executed in the general architecture of the IRC framework, and the specifics of the ICC module are described below.



Particles launched during the IRC process that reached the outlet at time t are traced back
directly to the IC timing or time 0, and their locations at the IC timing are shown in the bottom
maps in Figure 3d (referring to control points of time t). The downstream area of the control points
has shorter transportation time to arrive at the outlet (e.g., water difference $\Delta S_1$), and the upstream
area of the control points takes longer to get to the basin outlet (e.g., water difference $\Delta S_2$).
Similarly, soil moisture in the impacted area can greatly impact the size of $\Delta S_2$ and flow conditions
after the timing $t_2$. Assuming initial conditions are only impactful during the early period and late
recession of the hydrograph, which is supported by the fact that these events are flood-producing
events with large QPE uncertainties dominating the vicinity of peak flow, ICC is used for
hydrological windows outside the peak flow windows. Following the same notation (backward-
in-time) in the IRC framework (Eq. 18), $wd(t)$ is calculated as the flow volume difference
between observed and simulated streamflows for the time interval defined by $t$ and $t - \Delta t$. A
'band' of region can therefore be identified, that is, a region formed by control points of time $t$
and control points of time $t - \Delta t$. This 'band' is then referred to as the impacted area of initial soil
moisture for time $t$, meaning basin discharge between time $t - \Delta t$ and time $t$ is impacted by initial
soil moisture at the delineated impacted area. Finally, $wd(t)$ is then converted to soil moisture
content and added to initial soil moisture within the impacted area (i.e. the 'band') and the details
can be found in Liao and Barros (2025b).

**2.3.10 Hydrological Skill Metrics**
The Kling-Gupta Efficiency (KGE) is calculated using observed and simulated streamflow
statistics at observation resolution $\tau$ (here 15 minutes) in this work:





$$KGE_\tau = 1 - \sqrt{(r-1)^2 + (\frac{\sigma_{sim}}{\sigma_{obs}} - 1)^2 + (\frac{\mu_{sim}}{\mu_{obs}} - 1)^2}$$
(19)

where $r$ is the correlation between simulations and observatiosn, $\sigma_{obs}$ is the standard
deviation of observed discharge, $\sigma_{sim}$ is the simulated discharge standard deviation, $\mu sim$ and
$\mu obs$ represent the average simulated and observed streamflow values, respectively.
The relative volume error (EV) is the relative difference between simulated flood volume
and observed flood volume:
$$EV = \frac{V_{sim} - V_{obs}}{V_{obs}}$$
(20)

Where V stands for volume of the flood. An EV>0, and an EV<0 mean overestimation and
underestimation, respectively.
EPT refers to the error in peak flow timing between observations and simulations. For its
calculation, only the highest peak is selected for calculating EPT if more than one peak is present.
In this work, EPT is determined by considering the entire flood rising limb to account for the
steepness of the rising limb, specifically, both the flood starting timing and the maximum flood
timing from the flood front rising limb are used for calculating the EPT.
EPV or error in peak volume (Qmax, cubic meters per second) is a relative error calculated
using peak flows from observations and simulations, and the equation is below:
$$EPV = \frac{Qmax_{sim} - Qmax_{obs}}{Qmax_{obs}}$$
(21)






**2.3.11 Study Domain and Model Setup**
Twenty-eight headwater basins are selected in the Appalachians as illustrated in Figure 1,
with basin drainage area ranging from 50 km$^2$ to 500 km$^2$. It is demonstrated in Figure 4 that these
basins scatter across the entire Appalachians. For example, Basin01 and Basin30 are over 2,000
km apart, with diverse weather and climate regimes, and large differences in geomorphology and
hydrogeology.

<Figure 4 here please>

Soil-related parameters are downloaded from a global high-resolution (1 km) soil data
repository (Zhang et al., 2018). For each basin, the vertical hydraulic conductivity remains the
same for the entire soil column. The lateral hydraulic conductivity in the unsaturated zone was
assumed to be two to three orders of magnitude larger than the vertical conductivity in the shallow
soil layers, with higher values where the stone fraction in the soils is higher (Carlson, 2010; Freeze
and Cherry, 1979). The final scaling factors were obtained through simple sensitivity analysis to
match the curvature and slope of the observed subsurface runoff recession curves (e.g., Linsley et
al., 1982; Chen and Kumar, 2001; Yildiz and Barros, 2007), and scaling factors are finally
determined as: 1500, 150, 15 and 1.5 for layer 1 (0-10 cm below terrain surface), layer 2 (10-75
cm below terrain surface), layer 3 (75-200 cm below terrain surface) and layer 4 (2-20 m below
terrain surface), respectively. No parameter optimization is done in this work, as the primary focus
of this work is to develop a QPE dataset that can consistently close the water budget while



controlling uncertainties from other sources, largely advancing the understanding of QPE
uncertainties across climate, weather, and geomorphological regimes.

Flood-producing events have been selected for the 28 headwater basins for recent years

from January 2021 to April 2024. A qualified event is determined based on the observed peak
flow, which must surpass 95% of available flow measurements for each basin. The choice of 95%
is a compromise because 99% would yield too few events, while 90% would be too close to the
annual flood. Additionally, rainfall runoff response time must be shorter than or equal to 6 hours
to be qualified as a flash flood event. Only warm season precipitation events from 2021 to 2024
are finally considered. Here, the warm season is specifically defined as from April 1st to September
30th. Note: data quality control is enforced, and events with missing streamflow records are
discarded.

For the Cataloochee Creek Basin (Basin05), located in the SAM known to have

experienced multiple flash floods in the past (Tao and Barros, 2013 and 2014), Liao and Barros
(2023) created a Historical Flood Record database (HFR) that includes a large number of extreme
rainfall events from 2008 to 2017. The event selection criteria when developing HFR also use the
same 95% flow threshold method. The difference is that the HFR also includes multiple winter-
time liquid precipitation events that result in cold-season flash floods. In total, there are 54 warm-
season events for Basin 05 in HFR, and these events are also used to expand the study sample size
in this work.

To warm up the DCHM, a traditional spin-up approach is used with iterative runs for the

hydrological year of 2021 (from the end of April to the end of September), and it generally reaches
equilibrium after 3-5 iterations. Subsequently, DCHM is continuously running from the beginning
of October 2021 onwards, to derive initial conditions for events after September 30th, 2021.



During this spin-up process, no parameter calibration is involved. The initial conditions are extracted from the last iteration of spin up run, and the following model outputs generated after October 1st, 2021.

**2.4 Caveats**

In the entire study domain, rain gauges are only installed in the Southern Appalachians, specifically in the vicinity of the Cataloochee Creek Basin (Basin 05). However, the rest of the regions are not equipped by raingauge networks, and therefore, no rain gauge bias correction is done for those basins, and the downscaled original dataset StageIV (i.e., $STIV_D$) is used as input for the IRC method and hydrological simulations in this study.

As an important component of the IRC framework, the Lagrangian tracking algorithm is only implemented when hydrological window changes, rather than following model temporal resolution (i.e., 5 minutes), due to practical computational constraints. Additionally, we do not differentiate peak flow points and recession inflection points between simulations and observations when classifying hydrological flow regimes/windows, and consistently use observations delineate hydrological windows simply because 1) particle locations are inherently much more uncertain when simulation time is getting longer partially due to numerical truncation errors and grid-based abruptly-changing velocity fields used in the Lagrangian tracking algorithm, and 2) the computational costs of the tracking algorithm. Very short travel times (i.e., <15 minutes) are ignored because of temporal resolution restrictions from streamflow observations. A systematic use of 24 hours for event total duration is imposed in this work to reduce excessive tracking workload, which might be problematic for events with very long and heavy tails, though not common for flash flood events in headwater basins.





The IRC-ICC recursive framework allows us to quantify QPE uncertainties more
realistically by improving initial soil moisture estimation, and this framework is numerically
efficient in terms of reaching hydrological equilibrium state within 3-5 iterations. In this work, the
stable state of IRC-ICC is reached when the KGE changes are bound by 0.05.
**3. Results and Discussion**
**3.1 Rain gauge Bias Correction**
The climatologically corrected $STIV_{DBKC}$ fields have a significantly accurate diurnal cycle
compared to only event-scale bias-corrected $STIV_{DBK}$. This process is illustrated in Figure 5 for
one rain gauge from each side of the ridges (eastern side: left panel; western side: right panel) in
the Southern Appalachians.

<Figure 5 here please>

Original $StageIV_D$ show higher biases over the western ridges (e.g., right panel) for all hours of
day, illustrating the difficulties of capturing seeder-feeder enhancement of low-level precipitation
systems (Duan and Barros, 2017). Also, the mid-day dry bias has been a problem for radar
measurements in this region. (e.g., Barros and Arulraj, 2019). Results show that $StageIV_{DBKC}$
datasets capture precipitation climatology better with smaller missing detection errors compared
to original StageIV. Figure 6 shows the diurnal characteristics of the missing percipitataion for
two raingauge locations for winter season (January-February and March – JFM) using StageIV,
and this phenonemon is observed for both the $StageIV_D$ (black) and $StageIV_{DBK}$ (cyan). These
missing cases correspond to light rainfall that have small rainfall measurements at rain gauge



locations (< 1.5 mm/hr, bottom row). After applying precipitation climatology corrections, the
missing issue in StageIV$_{DBK}$ is significantly alleviated and much better results are shown in
StageIV$_{DBKC}$ fields (green).

<Figure 6 here please>

The seasonal HSS, TS, and RMSE of STIV$_{DBKC}$ are significantly better than those of
STIV$_D$ throughout the day using 10-year averages (Figure 7a). It is worth noting with increasing
precipitation rate threshold (Figure 7b), threat score does not show decreasing trend, meaning
raingauge bias correction for heavy rainfall events works well. Figure 7c shows RMSE
performance conditional on rain rate at diurnal and seasonal scales. Overall, the RMSE is generally
less than 0.1 mm/hr except in the cold-season morning and late afternoon, which can be partially
attributed to snow events because these raingauges are not heated.

<Figure 7 here please>

**3.2 Hydrologic Correction**
The coupled IRC-ICC was originally developed and applied in Basin 05, the Cataloochee Creek
Basin, and an example showing the results from iterations is demonstrated in Figure 8. The
notation follows the definition in Figure 3. Note that the STIV$_{DBKC}$ data derived in Section 3.1 are

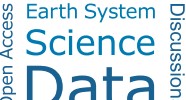

further downscaled to 250m and used for hydrological simulations in this section. For all other
basins (except Basin05), rain gauges are not available, and STIV$_D$ data are used instead.

<Figure 8 here please>

It is demonstrated that IRC-ICC produces stable results after about 3 to 4 iterations without
significant oscillations for this specific extreme flood event. In general, for less significant events,
IRC-ICC reaches equilibrium faster (merely three iterations), providing fast and convergent
corrections. As explained earlier, the equilibrium state is reached and thus IRC-ICC is stopped
when oscillations in simulated KGE are within 0.05, and then IRC-ICC is stopped immediately.
This study suggests that for most events, three iterations is a good rule of thumb.   The difference
between the initial 4D (x, y, z, t) rainfall forcing and the final result of the IRC-ICC is the general
IRC correction.

**3.2.1 Systematic Application of IRC-ICC**

The IRC-ICC is systematically executed in the 28 basins located in the Appalachians for

225 events, and examples are displayed in Figure 9.

<Figure 9 here please>

Simulated streamflows generally have better performances in the Northern and Southern

Appalachian Mountains (NAM, SAM) compared to the Central (CAM). Specifically, in the Karst
region along the interstate border of Virginia and West Virginia in the CAM,  for Basins 13 and
14, where there are numerous caverns and natural tunnels facilitating fast subsurface flow response,
that is, sinking and subterranean streams (https://www.dcr.virginia.gov/natural-
heritage/vacavetrail and https://docslib.org/doc/2284608/west-virginia-tax-districts-containing-
karst-terrain). The current version of the DCHM does not have a specific module designed for
karst geology and karst hydrological processes. Thus, the IRC-ICC results in these locations are
impacted by model structural uncertainty.  Here, the advantage of not calibrating model parameters
becomes apparent. It would be possible to calibrate model parameters to improve model
simulations; however, the physical basis and transferability of the IRC-ICC results would be
compromised. The 10 events in Basins 13 and 14 are therefore discarded (example: Figure A3).
This point of discussion is highlighted here to reinforce the value of the data set presented in this
manuscript for applications with other hydrologic models, including model calibration, where
model structural uncertainty is not a primary concern at resolved scales.

Event 2021-06-10 in Basin 19 (see Figure A3) is an example of an event with a complex

hydrograph (e.g., multiple minor flood peaks around one major flood peak) that requires more
hydrological windows (see Figure 3). Subtle changes in the hydrograph shape could be indicative
of spatial shifts in runoff production from one tributary to another following the track of storm
cells over the basin.  Indeed, depending on the weather system and regional topography, the travel
velocity of such cells and their life-cycle may require finer spatial and temporal resolution both
for the hydrological model and for the tracking algorithm to capture changes in the spatial structure
of precipitation, especially in the case of summer thunderstorms. For the systematic production of



this data set, a 5-window IRC-ICC framework was applied, including a pre-rising-point segment,
rising limb, early recession, and late recession (separated by the recession inflection point).

**3.2.2 IRC and IRC-ICC Precipitation Corrections**
Accumulated rainfall totals per rainfall event are calculated for both the IRC-only product
and post IRC-ICC products. Subsequently, these rainfall totals are directly compared against
original product $STIV_{DBKC}$. Examples are shown in Figure 10, categorized by seasons in the
Cataloochee Creek Basin (Basin05). Again, the warm season is defined as April 1st to September
30th, and the remaining events are defined as the cold season, with only liquid precipitation events
studied in this work.

<Figure 10 here please>

The original QPE (**a1** and **b1**) shows abrupt changes in rainfall intensity, which is a
common issue of radar observations at high spatial resolution. On the contrary, the IRC-corrected
precipitation maps demonstrate precipitation features aligning with landform, showing strong
spatial precipitation gradients along ridges and adjacent valleys (examples are listed in Figure A2).
The spatial correlation between orographic precipitation and topography is observed across all
mountain ranges, including the Appalachians (e.g., Konrad II, 1994; Smith et al., 2011; Wolvin et
al., 2024). Note the dark blue colors in Figure 10 corresponding to very low precipitation near the
basin outlet are an artifact of the IRC tied to very short travel times that cannot be fully resolved
even at fine scales of 250m and 5minutes. However, these artifacts are much reduced for the IRC-

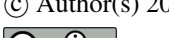



ICC due to the reduction of uncertainty in initial conditions, as shown for the 2009-10-14, 2009-
04-20, and 2013-04-12 events because of overall basin-wide travel time improvement. It is worth
noting that these three events are relatively mild events, indicating a larger impact of IC on
relatively less extreme events because of the critical role of IC in runoff generation mechanisms
and travel times distributions. Thus, the extreme event precipitation product obtained from IRC-
ICC is the data set recommended for applications with other hydrologic models.

### 3.2.3 Precipitation and Hydrologic Statistics


Event-total precipitation maps are calculated for each basin and event, and basin-scale
precipitation statistics (e.g., mean and standard deviation) are derived for each event-total
precipitation map. These statistics are plotted in Figure 11, and subregions are separated by vertical
black lines. Basins 01 to 11 are located in the SAM, Basins 12 to 20 are located in the CAM, and
Basins 21 to 30 are located in the NAM. Basins 13 and 14 are not included in the statistics.
<Figure 11 here please>
It is clearly demonstrated that the change in the mean (i.e., basin-averaged event total QPE)
is relatively small (from 36.10mm to 38.07mm) compared to the change in the standard deviation
(from 6.63mm to 14.08mm) after the application of IRC-ICC. The small standard deviation of the
original QPE suggests that the original QPE data are spatially tightly clustered with low variability
(see Figure 10a for boxy rainfall features), while the larger standard deviation post-IRC-ICC
indicates spatial variability is enhanced, which is highlighted by the terrain-aligned precipitation
features in Figure 10c. The relatively small change in the mean indicates that the original input
precipitation (i.e., StageIV$_{DBKC}$ for Basin 05, and StageIV$_D$ for the remaining basins) does not





contain significant unconditional systematic biases across basins and events, which would lead to
consistent positive or negative flood volume errors.  As an exception, it is worth noting that the
standard deviation of Basin 05 events does not change significantly after the IRC-ICC compared
to other basins and events because rain gauge corrections from the IPHEx network are employed
in Basin 05 but not anywhere else. It can never be overly emphasized that even after rain gauge
bias correction, essentially a point-scale correction method, the resulting flood hydrograph exhibits
significant water budget closure errors (see Figure 12 for more discussion)  on account of the high
heterogeneous nature of QPE in complex terrain.

The hydrologic statistics described in Table 1 using all studied events are plotted in Figure

12.

<Figure 12 here please>

Figure 12 shows that the median KGE across events is improved from 0.36, 0.39, 0.27 to

0.89, 0.74, 0.84 for SAM, CAM, and NAM, respectively. It should be pointed out that QPE
changes for Basin 05 events (event numbers 55 to 108) are important for improving water budget
closure, albeit small in magnitude compared to other events in other basins, as shown in Figure 11
and 12, and yet critical to capture the complex precipitation heterogeneity in complex terrain to
close the water budget. The results for Basin 05 illustrate the limitations of rain gauge-based bias
corrections in complex terrain in general. The relatively small improvement shown in the CAM is
partially attributed to the fact that DCHM does not have a proper representation of subterranean
rivers in karst terrain, causing large baseflow errors during hydrograph recession and thus low
KGE values. Nevertheless, for flash flood applications, peak flow magnitude, flood flow timing,
and event flow volume are the most important forecast objectives, corresponding to the 2nd, 3rd,
and 4th horizontal panels in Figure 12. Overall, flood volume error (EV) is controlled within ±10%





for over 90% of the studied events (the 2$^{nd}$ panel), with the median EV error being less than 5% in
the SAM and NAM after IRC-ICC corrections. Flood peak volume (the 3$^{rd}$ panel) is generally
controlled within 20%, which is very good for extreme events in regions without ground-based
observations except for radars placed far away. This is demonstrated by Tropical Storm Fred on
2021-08-17: an event that caused floods in multiple SAM basins, caused five deaths, and resulted
in an economic loss of more than 1 billion dollars. Note the KGE for this event is improved to 0.9,
and peak timing errors are <30 minutes using IRC-ICC. Timing errors (shown in the 4$^{th}$ subplot)
are bounded by ±60 minutes for the major of the events for post IRC-ICC datasets, though some
outliers exist potentially due to complex antecedent land surface physics (e.g., rain on snow) for
April events, particularly in the CAM and NAM.
Events associated with significant timing errors (more than ±90 minutes) are investigated
in detail.  These include the 2023-07-08 event (event number 185) for Basin 27, which is located
in New Hampshire (the estimated flood front occurs too early by 2.5 hours). This was a localized
summer thunderstorm event, only taking 30 minutes to reach its peak flow. The fast changes in the
hydrological regime require much more windows than the current classic 5-window settings used
in the IRC-ICC framework. The event on 2022-05-27 (event number 118) in Basin16 located in
West Virginia is characterized by a slow rising limb. Note Basin16 is partially located in a complex
region with karst features (e.g., sink holes) in the Greenbrier-river valley. Finally, the event 2021-
09-22, a complex rainfall system characterized by multiple rain cells passing through the Basin 19
quickly (event number 133), requiring smaller hydrological windows to capture highly variable
rainfall-runoff responses than the 5-window default IRC-ICC architecture: baseflow segment, pre-
rising segment, flood rising limb, early and late recession.





Overall, large improvements in QPE are achieved, resulting in hydrological improvements
in aspects of peak magnitude, flood total volume and flood front timing. Due to the dependence of
IRC-ICC on travel time distributions, it cannot be used when precipitation is missing or there are
severe timing errors because of the lack of water travel time trajectories to distribute corrections.
From a practical point of view, the QPE IRC-ICC correction is in nature a type of space-time bias
correction. The improved QPE data facilitates the development of QPE error models, which is
demonstrated by the same authors (e.g., Liao and Barros, 2023), providing a path towards
correcting remote-sensing products to support hydrometeorological studies and advancing the
calibration of hydrological models with significantly less forcing uncertainty.

**683    3.2.4 Independent Verification**

As mentioned in the introduction, precipitation measurements are limited in the Appalachians
except for the IPHEx rain gauge network (Figure 1). Currently, the NEXRAD radar network
remains the widely used precipitation monitoring system in this region in spite of well-documented
low radar quality coverage over radar gaps in the mountains. The Multi-Radar/Multi-Sensor
(MRMS) product (Zhang et al., 2016), which is developed using NEXRAD radar measurements
similar to StageIV, is created at 1km resolution and is used here for independent verification.
First, original MRMS data are downscaled to the same resolution as StageIV$_D$ datasets (250m,
5min) and used as inputs for DCHM. Hydrological simulations in this section are using the same
model configuration and initial model states for the purpose of a meaningful comparison, including
the following datasets: MRMS$_D$, StageIV$_D$, and IRC-ICC StageIV$_D$ as shown in Figure 13. Figure
13a shows that MRMS and StageIV QPE have similar results. Second, the IRC-ICC StageIV$_D$



have generally a good agreement with $MRMS_D$ similar to $StageIV_D$. However, for some cases,
where rainfall is dramatically underestimated by the radar system and KGE values are low, IRC-
ICC is shown to provide effective corrections. Otherwise, the IRC-ICC generates physically
constrained corrections spatially (see Figure 10), achieving high KGE values for flood simulations.
Figure 13b shows the histogram of the KGE values across different rainfall products for all events.
Overall, simulated streamflows using $MRMS_D$ and $StageIV_D$ exhibit similar hydrologic
performance (the median KGE across events is close to 0.20), on the contrary, post-IRC-ICC
$StageIV_D$ produce flood simulations with a median KGE above 0.80.

## 4. Discussion and Future Work

Limitations in this study stem mainly from computational constraints rather than
methodology. A default 24-hour flood duration window is imposed, implying that for long-lasting
floods, due to significant slow interflow and baseflow contributions, are not considered. The
current version of the IRC-ICC framework was built to support flash flood studies and only targets
shallow subsurface moisture transport, given the critical importance of shallow soil moisture on
the regulation of flood generation and propagation in steep terrain. It is worth noting that for long-
lasting rainfall events or regions with relatively flat terrain, slow interflows would become more
important in terms of regulating flood timing, flood volume, and post IRC-ICC QPE.
While the IRC results could be further optimized if carried out at the same frequency as
the model resolution, therefore eliminating any artifacts due to inadequate sampling and updating
of travel time distributions, and while there is room to improve the IRC-ICC framework through
improved model physics and resolution, utilizing 3D velocity fields to capture the full travel time



distributions, and using different models to generate IRC ensembles. to test and calibrate
hydrologic models for an intercomparison study, advancing flood forecasting skill, and to support
emergency management response.

## 5. Data Availability Statement
The StageIV-IRC dataset at 250 m 5-minute resolution for 26 basins and 215 events is
available at: https://doi.org/10.5281/zenodo.14028866. (Liao and Barros, 2025c), excluding
Basin 13 and 14 based on previous discussion. Associated geographic documentation of the
selected basins is also provided via the same link. Initial soil moisture distributions for the studied
events are also available in the same Zenodo repository.

## 6. Conclusion
QPE has been an enduring challenge in hydrology, particularly in complex terrain. Ground-
based radar QPE is plagued with uncertainties from multiple sources, while rain gauge networks
are scarce and suffer from the lack of representativeness in the mountains. To address this grand
challenge, we develop a series of corrections from point-scale to watershed-scale encompassing
event bias, climatology, and water budget closure: the IRC-ICC framework. To our knowledge,
this is the first QPE dataset that meets standard statistical evaluations against point-based
measurements where available and meets water budget closure at flood-event scale, consistent
with nonlinear rainfall-runoff processes in headwater basins, and achieves superior hydrological
performance at sub-hourly.





The IRC-ICC framework is successfully adopted in 26 mountainous basins (excluding the
basins that are heavily overlapped with Karst terrain) in the Appalachians for 215 events with
robust success, yielding substantial improvements of streamflow simulation, particularly in terms
of flood volume and timing. The tracking algorithm in the IRC-ICC framework is only updated
when shifting from one hydrological window to another, but not every time step. With enough
computational resources, post-IRC-ICC QPE data should further improve by capturing transient
travel time distributions between model time steps.
When using the StageIV-IRC product, flood timing errors are controlled with one hour for
90% of events, compared to less than 20% when using original StageIV, while the median KGE
improved from 0.34 to 0.86 across the events. This change in KGE is achieved by significant
changes in the space-time variance of precipitation that in turn impacts the space-time variability
of rainfall-runoff processes. Results illustrate the importance of initial conditions for less severe
rainfall events, particularly during the beginning of the event, which influences subsequent
streamflow simulations. It should be emphasized that physical parameters are not calibrated for
any precipitation event in any basin in this work. This physics-based IRC-ICC framework can
capture the fundamental physics involved in flash flood events: essentially the fast rainfall-runoff
responses in surface and shallow subsurface layers; therefore, skillful hydrologic prediction is
achieved without model calibration. Instead, the focus is on getting the forcing right.
The IRC-ICC is a general framework that can be incorporated into any distributed
hydrological model. Thus, the StageIV-IRC dataset also enables meaningful intercomparison
among different radar QPE datasets, providing physics insights into QPE error structure from a
water budget closure perspective, toward improving radar retrievals and to characterize radar-
specific errors related to radar operations at high spatial resolution in the mountains. The

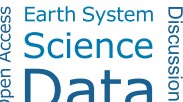

demonstrated success of StageIV-IRC in ungauged basins strongly supports the use of IRC-ICC
in mountainous regions worldwide, where rain gauges are generally not available. Further, this
dataset can be utilized as a reference for building machine learning models (or even deep-learning
models when the number of studied precipitation events is expanded) that can learn the QPE
uncertainties conditional on time of day, weather, climate and geomorphological regimes for both
radar QPE analysis and forecasts, advancing the understanding and quantification of orographic
precipitation uncertainty at high resolution across global mountains.
## 7. Appendix A
The detailed distribution process of water difference (wd) is illustrated in Figure A1 following
Section 2.3.8.

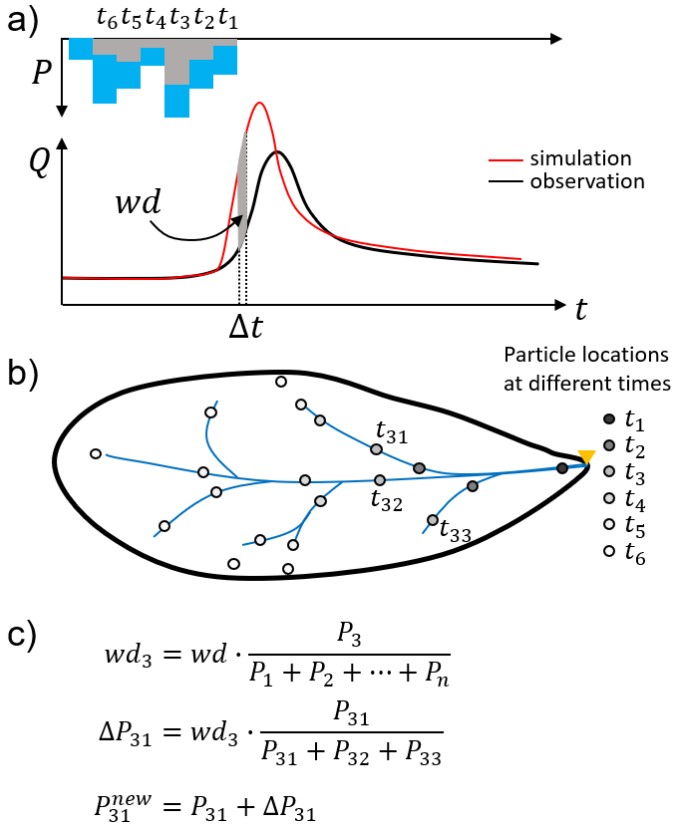


**Figure A1 -** Schematic depiction of the IRC framework and key mathematical equations. Panel

**(a)** illustrates the nonlinear relationship between streamflow and precipitation, where $wd$

represents the residual between discharge simulations and observations at the basin outlet. The

variation of precipitation in the basin as a function of time is shown by the basin hyetograph in

blue. The hyetograph time series (blue) spans the duration of the precipitation event between $t_1$

to $t_n$. In gray is the hyetograph over the area of interest for panels (b) and (c). To map the

streamlines, water particles are launched every time step and their trajectory to the outlet is tracked

and saved. Panel **(b)** shows the source areas of water particles launched at various time steps

$(t_1, t_2, \dots t_6 \dots)$ from all locations where runoff is produced, and the particles are tracked until they

eventually reach the basin outlet. The streamlines of particles that reach the outlet at the same time





are used to distribute the residuals backwards to the runoff source areas where the particles were
originally launched (e.g., the three particles $t_{31}, t_{32}$, and $t_{33}$ that reach the basin outlet at time $t_3$).
Panel **(c)** shows the algorithm to calculate the rainfall bias correction at location $t_{31}$ due to the
residual $wd_3$ at time $t_3$. $P_i$ is basin averaged rainfall at time $t_i$, and $wd_3$ is the runoff volume to
be corrected at time step $t_3$. $\Delta P_{31}$ is the precipitation correction for pixel $t_{31}$, and precipitation
amount at pixel $t_{31}$ before and after IRC are denoted by $P_{31}$ and $P_{31}^{new}$. This figure is adapted from
Liao and Barros (2025b).

A zoom in map of the Southern Appalachians is plotted associated with DEM maps of other basins.
A complete set of maps for each individual basin can be requested. Note, the rain gauges used in
this study are plotted in Figure 1, and they are primarily near Basin05.

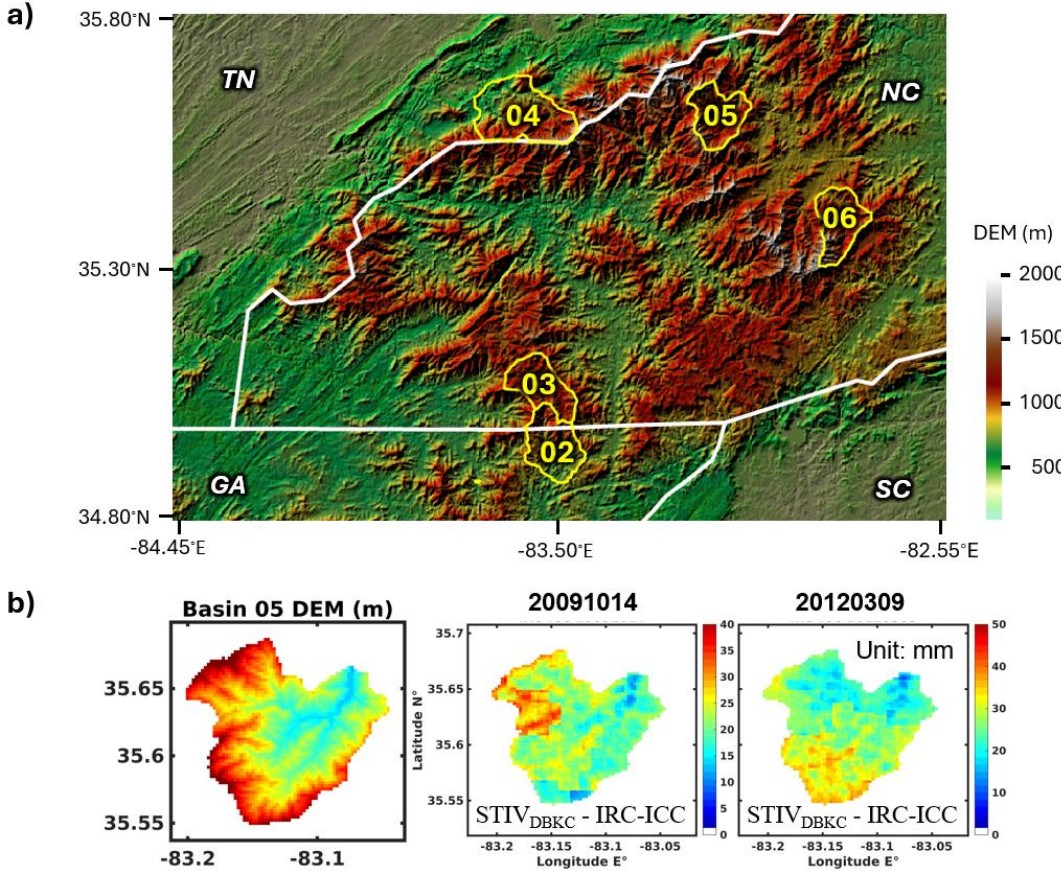


**Figure A2** – A zoom-in map of the Figure 4 for watersheds in the Southern Appalachians (Panel

a). The DEM map and examples of rainfall event accumulation of Basin 05 (Panel b) to show

rainfall alignment with topography.







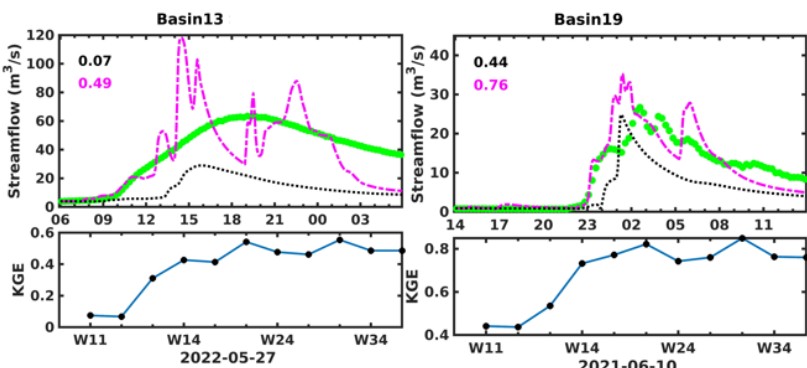


**Figure A3** – Examples of the coupled IRC-ICC framework application in Basin 13 and Basin 19
for discussion in the manuscript. Basin 13 is located in Karst terrain, while the event in Basin 19
is an example with a complex hydrograph.

**CREDIT AUTHOR STATEMENT**
M. Liao: Methodology, Data curation, Writing - original draft, Investigation. A. P. Barros:
Conceptualization, Methodology, Writing - review & editing, Supervision, Project administration,
Funding acquisition.
**COMPETING INTERESTS**
The authors declare there are no competing interests.
**ACKNOWLEDGMENTS**
The work was supported by NASA Earth System Science Fellowship associated with the first
author and supported by a joint effort from NASA grant 80NSSC19K0685 and a grant from the
IBM Accelerator program with the second author.



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





LIST OF TABLES





**Table 1** – Raingauge index and exact locations as illustrated in Figure 1. Two rain gauges highlighted in bold font are installed at Purchase Knob, a supersite in the inner mountain region. Locations equipped with more than one raingauge (collocated) are shaded in grey color, and these collocated raingauges generally differ in tipping sizes. This table is adapted from Liao and Barros (2019).

| NO. | Site ID. | Latitude | Longitude | Elevation (m) |
|---|---|---|---|---|
| 01 | RG 001 | 35.398 | -82.913 | 1156 |
| 02 | RG 002 | 35.417 | -82.971 | 1731 |
| 03 | RG 003 | 35.384 | -82.916 | 1609 |
| 04 | RG 004 | 35.368 | -82.990 | 1922 |
| 05 | RG 005 | 35.408 | -82.964 | 1520 |
| 06 | RG 008 | 35.382 | -82.973 | 1737 |
| 07 | RG 010 | 35.456 | -82.946 | 1478 |
| 08 | RG 100 | 35.586 | -83.072 | 1495 |
| 09 | RG 100T | 35.587 | -83.064 | 1485 |
| 10 | RG 101 | 35.575 | -83.088 | 1520 |
| 11 | RG 102 | 35.563 | -83.103 | 1635 |
| 12 | RG 103 | 35.553 | -83.117 | 1688 |
| 13 | RG 104 | 35.554 | -83.088 | 1584 |
| 14 | RG 106 | 35.432 | -83.029 | 1210 |
| 15 | RG 109 | 35.495 | -83.040 | 1500 |
| 16 | RG 110 | 35.548 | -83.148 | 1563 |
| 17 | RG 300 | 35.726 | -83.216 | 1558 |
| 18 | RG 301 | 35.705 | -83.255 | 2003 |
| 19 | RG 302 | 35.721 | -83.246 | 1860 |
| **20** | **RG 303PK** | **35.586** | **-83.072** | **1495** |
| **21** | **RG 303S** | **35.762** | **-83.162** | **1490** |
| 22 | RG 304 | 35.670 | -83.182 | 1820 |
| 23 | RG 305 | 35.691 | -83.131 | 1630 |
| 24 | RG 306 | 35.745 | -83.171 | 1536 |
| 25 | RG 307 | 35.651 | -83.199 | 1624 |
| 26 | RG 308 | 35.730 | -83.182 | 1471 |
| 27 | RG 309 | 35.682 | -83.150 | 1604 |
| 28 | RG 310 | 35.702 | -83.122 | 1756 |
| 29 | RG 311 | 35.765 | -83.140 | 1036 |
| 30 | RG 400 | 35.702 | -83.122 | 1756 |
| 31 | RG 401 | 35.651 | -83.199 | 1624 |
| 32 | RG 402 | 35.721 | -83.246 | 1860 |
| 33 | RG 403 | 35.517 | -83.101 | 925 |
| 34 | RG 407 | 35.517 | -83.101 | 925 |



**Table 2**: Hydrologic skills used in this work.

| Notation | Information | Reference |
|---|---|---|
| KGE | Kling-Gupta efficiency | Eq. (19) /Gupta et al. (2009) |
| EV | Relative error in flood volume | Eq. (20) |
| EPT | Error in peak flow timing | Flood front timing differences |
| EPV | Relative Error in maximum flow rate | Eq. (21) |





Table 3 – Information table for selected basins and corresponding streamflow gauges used in this
work. This table is adapted from Liao and Barros (2025b).

| Basin index | USGS Gauge ID | Drainage area (km$^2$) | Basin highest elevation (m) | Basin relief (m) | Location |
|---|---|---|---|---|---|
| 1 | 3544970 | 118.7 | 1442 | 847 | GA |
| 2 | 2178400 | 176.1 | 1629 | 1051 | GA |
| 3 | 3504000 | 149.9 | 1667 | 1032 | NC |
| 4 | 3497300 | 317.6 | 1999 | 1651 | TN |
| 5 | 3460000 | 148.1 | 1879 | 1174 | NC |
| 6 | 3456500 | 152.8 | 1873 | 1157 | NC |
| 8 | 344894205 | 41.3 | 1995 | 1221 | NC |
| 9 | 3463300 | 134.3 | 1989 | 1425 | NC |
| 10 | 3400500 | 234.7 | 1257 | 1257 | KY |
| 11 | 3479000 | 283.3 | 1772 | 1216 | NC |
| 13 | 3182700 | 447.3 | 1111 | 717 | WV |
| 14 | 2011460 | 194.4 | 1388 | 763 | VA |
| 15 | 1620500 | 54.5 | 1321 | 712 | VA |
| 16 | 3180500 | 426.8 | 1416 | 621 | WV |
| 17 | 3068800 | 437.1 | 1471 | 908 | WV |
| 18 | 1595000 | 234.8 | 1230 | 560 | MD |
| 19 | 1595300 | 130.3 | 1069 | 712 | WV |
| 20 | 1544500 | 445.9 | 765 | 457 | PA |
| 21 | 1422747 | 81.4 | 766 | 394 | NY |
| 22 | 1415000 | 106.8 | 1019 | 636 | NY |
| 23 | 1413398 | 152.8 | 1094 | 754 | NY |
| 24 | 13621955 | 41.7 | 1074 | 717 | NY |
| 25 | 1421610 | 51.3 | 970 | 497 | NY |
| 26 | 1074520 | 389.4 | 1582 | 1582 | NH |
| 27 | 10642505 | 294.9 | 1895 | 1693 | NH |
| 28 | 1137500 | 300.3 | 1894 | 1546 | NH |
| 29 | 1133000 | 183.2 | 975 | 719 | VT |
| 30 | 1055000 | 334.1 | 1143 | 975 | MAINE |





## LIST OF FIGURES

**Figure 1 -** Map of IPHEx (Barros et al., 2014) ground-based observations in the Southern Appalachians. Raingauge is denoted as a character string starting with three-digit number potentially followed by extra letters; locations started with a letter P represent disdrometers. The basic information regarding these stations is listed in Table 1. This figure is adapted from Liao and Barros (2019).

**Figure 2 –** Workflow to generate the product STIV$_{DBKC}$.

**Figure 3** – An illustration of the structure of IRC, ICC and the coupled IRC-ICC framework including **a)** the residual hydrograph between the observed and simulated discharge, with the discharge water difference *wd(t)* being distributed across the time window T; **b)** Example of travel time distribution TT(t) and map (inset) illustrating a hypothetical distribution of runoff source areas (in red, ns=3) with travel time $x_2$ contributing to streamflow at time t, meaning that at time t-$x_2$ there are three pixels (ns=3) generating runoff that reaches the outlet at time t. T is the time window over which runoff source areas with TT < T are mapped and the inverse rainfall correction (IRC) are applied; **c)** Example of IRC windows guided by timescales of dominant hydrological processes. The first window solely covers the initial streamflow conditions before the target event. The second window depicts the early rising limb of the hydrograph. The third window captures the steep rising limb of the hydrograph until it reaches the peak flow. The fourth and fifth windows correspond to interflow-dominant and baseflow-dominant stages of the recession curve respectively, separated by the recession inflection point; **d)** A schematic drawing that shows different characteristic timings in a hydrograph with the implementation of the Initial Condition Correction (ICC) strategy. Specifically, $T_{r*}$ and $T_r$ represent the timing of flood front in simulations and observations, respectively. $T_p$ is the timing of observed maximum flood. The inflection point of the recession curve of the observations is denoted as $T_I$. Flow differences at $t_1$ and $t_2$ are denoted as $\Delta S_1$ and $\Delta S_2$ respectively for the purpose of discussion. P, Q and IC represent precipitation, flow discharge and initial condition, respectively; **e)** The implemented framework in this work consisting of ICC and IRC. This figure is adapted from Liao and Barros (2022, 2025b).

**Figure 4** – Map of the Continental United States (CONUS) and headwater basins studied in this work. Basin information is available in Table 3. Sub-regions are delineated as the following for discussion purposes only: Northern, Central and Southern Appalachian Mountains (NAM, Basin 21-30; CAM, Basin 13-20; SAM, Basin 01-11). This figure is adapted from Liao and Barros (2025b).

**Figure 5** - Examples of raingauge measurements showing the diurnal cycle of different seasons at different locations: Left panel – raingauge RG008 located in the eastern ridges for the Summer (JAS: July-August-September) season. Right panel – raingauge RG302 located in the western ridges for the Spring (AMJ; April-May-June) season. Rain gauge measurements (blue); StageIV$_{DBK}$ (black); StageIV$_{DBKC}$ (green). This figure is from Liao and Barros (2019).

**Figure 6** –Top row – The diurnal cycle of missing precipitation at RG003 (Eastern ridges) and RG103 (Inner regions) for January-February-March (JFM) using various products. Bottom row- corresponding rain gauge climatology (blue). StageIV$_D$ (black); StageIV$_{DBK}$ (cyan); StageIV$_{DBKC}$ (green). This figure is from Liao and Barros (2019).





**Figure 7** – Statistical evaluation summary for winter precipitation (JFM, January, February, and March): a) Diurnal cycle of mean HSS and TS statistics including all rain gauges calculated using all data from 2008 to 2017: $STIV_D$ (black) and $STIV_{DBKC}$ (green); b) HSS and TS statistics calculated using different rain rate thresholds over the same 10-year period; c) Diurnal cycle of rain rate RMSE at seasonal-scale, and its dependence on observed rainfall rate. This figure is from Liao and Barros (2019).

**Figure 8** – The IRC-ICC performance in Basin05 as an example for the 2017-10-23 event (Basin05: Cataloochee Creek Basin, NC). This event is part of the 2017 Hurricane Nate. This figure contains **a)** hydrological responses when precipitation forcing is the $STIV_{DBKC}$. The dashed rectangular plot consisting of intermediate results including each iteration from the IRC-ICC framework (Figure 3). **b)** the hydrological equilibrium of the IRC-ICC after 5 iterations. This figure is adapted from Liao and Barros (2025b).

**Figure 9** – The IRC-ICC performance for different subregions, include **a)** 3 events from the Southern Appalachians; **b)** 3 events from the Central Appalachians; and **c)** 3 events from the Northern Appalachians. The IRC-ICC KGE evolution plots from iterations are included below the hydrographs. The black and pink line are from the original $STIV_D$ and the IRC-ICC equilibrium state ($STIV_D^{IRC*}$), respectively, and KGE values are displayed as colored numbers in the top left corners. This figure is adapted from Liao and Barros (2025b).

**Figure 10** – Event total precipitation maps for three cold season events (a) and three warm season events (b). Each column represents one event, and each row represents one precipitation product: $STIV_{DBKC}$, $STIV_{DBKC}^{IRC*}$ from IRC-only framework, and $STIV_{DBKC}^{IRC*}$ from the coupled IRC-ICC framework. This figure is adapted from Liao and Barros (2025b).

**Figure 11** – Summary charts of precipitation statistics for all event-total precipitation maps. Basin precipitation average and standard deviation for each event are represented by circles and triangles in the top and bottom panel, respectively. Each panel consists of 3 sub-regions by vertical black lines: the Southern Appalachian Mountains, Central Appalachian Mountains, and Northern Appalachian Mountains. The list of events in Basin 05 (with event number ranging from 55 to 108) in the SAM is highlighted by a blue rectangle for further discussion in the text. The average values of all events for both the mean and the standard deviation are calculated and shown in the top right corner. Black color and pink color represent pre and post IRC-ICC QPE statistics, respectively.

**Figure 12** – Summary of hydrologic skills. Green dashed lines and associated uncertainty envelopes are only for visual illustration. Hydrologic statistics are explained in Table 2. Pink and black scatters (each scatter represent one event) represent IRC-ICC, and baseline outputs, respectively. Each horizontal panel is split into 3 subsections by vertical black lines representing the 3 subregions. Histograms graphs on the right hand side are provided for a summary view. This figure is adapted from Liao and Barros (2025b).

**Figure 13** – **a)** Event total QPE plots for various QPE datasets conditional on seasons and KGE values; **b)** KGE distributions across events using different QPE datasets. This figure is adapted from Liao and Barros (2025b).



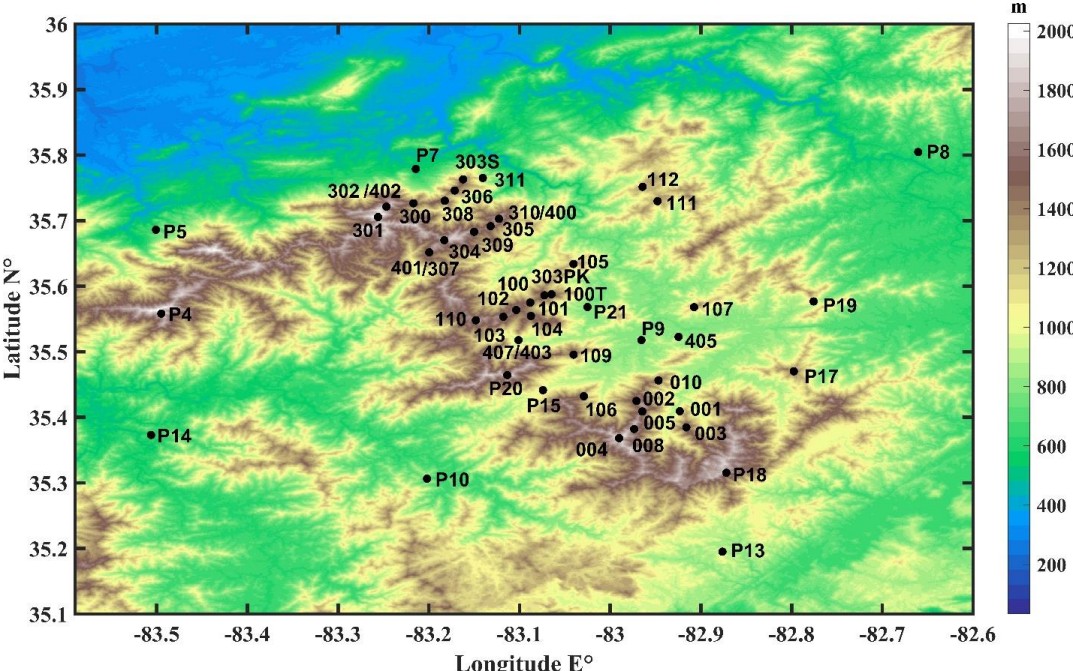

**Figure 1 -** Map of IPHEx (Barros et al., 2014) ground-based observations in the Southern
Appalachians. Raingauge is denoted as a character string starting with three-digit number
potentially followed by extra letters; locations started with a letter P represent disdrometers. The
basic information regarding these stations is listed in Table 1. This figure is adapted from Liao and
Barros (2019).



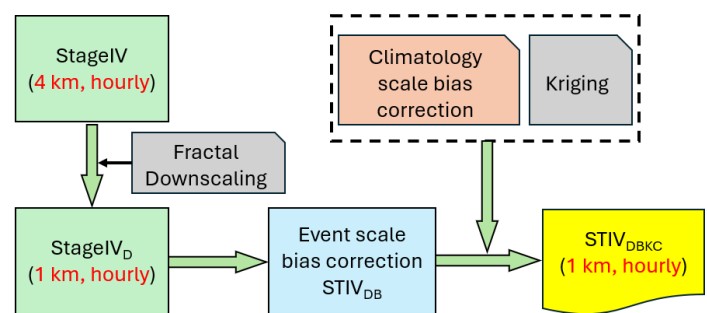

**Figure 2** – Workflow to generate the product STIV$_{DBKC}$.

1267



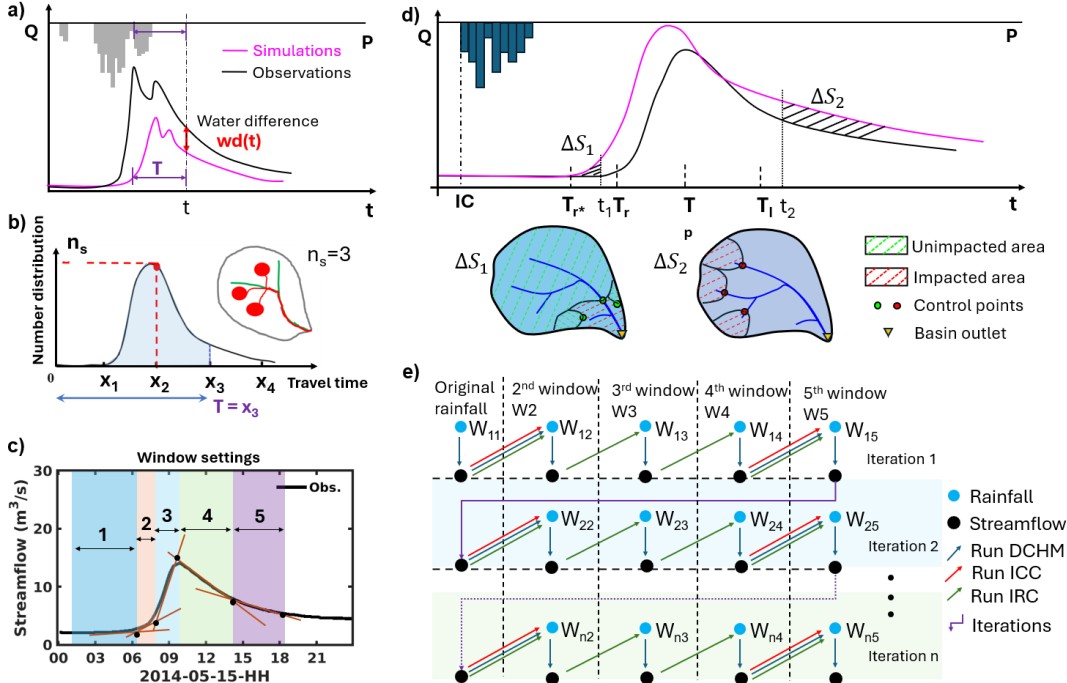

**Figure 3** – An illustration of the structure of IRC, ICC and the coupled IRC-ICC framework including **a)** the residual hydrograph between the observed and simulated discharge, with the discharge water difference *wd(t)* being distributed across the time window T; **b)** Example of travel time distribution TT(t) and map (inset) illustrating a hypothetical distribution of runoff source areas (in red, ns=3) with travel time $x_2$ contributing to streamflow at time t, meaning that at time $t-x_2$ there are three pixels (ns=3) generating runoff that reaches the outlet at time t. T is the time window over which runoff source areas with TT < T are mapped and the inverse rainfall correction (IRC) are applied; **c)** Example of IRC windows guided by timescales of dominant hydrological processes. The first window solely covers the initial streamflow conditions before the target event. The second window depicts the early rising limb of the hydrograph. The third window captures the steep rising limb of the hydrograph until it reaches the peak flow. The fourth and fifth windows correspond to interflow-dominant and baseflow-dominant stages of the recession curve respectively, separated by the recession inflection point; **d)** A schematic drawing that shows different characteristic timings in a hydrograph with the implementation of the Initial Condition Correction (ICC) strategy. Specifically, $T_{r*}$ and $T_r$ represent the timing of flood front in simulations and observations, respectively. $T_p$ is the timing of observed maximum flood. The inflection point of the recession curve of the observations is denoted as $T_I$. Flow differences at $t_1$ and $t_2$ are denoted as $\Delta S_1$ and $\Delta S_2$ respectively for the purpose of discussion. P, Q and IC represent precipitation, flow discharge and initial condition, respectively; **e)** The implemented framework in this work consisting of ICC and IRC. This figure is adapted from Liao and Barros (2022, 2025b).



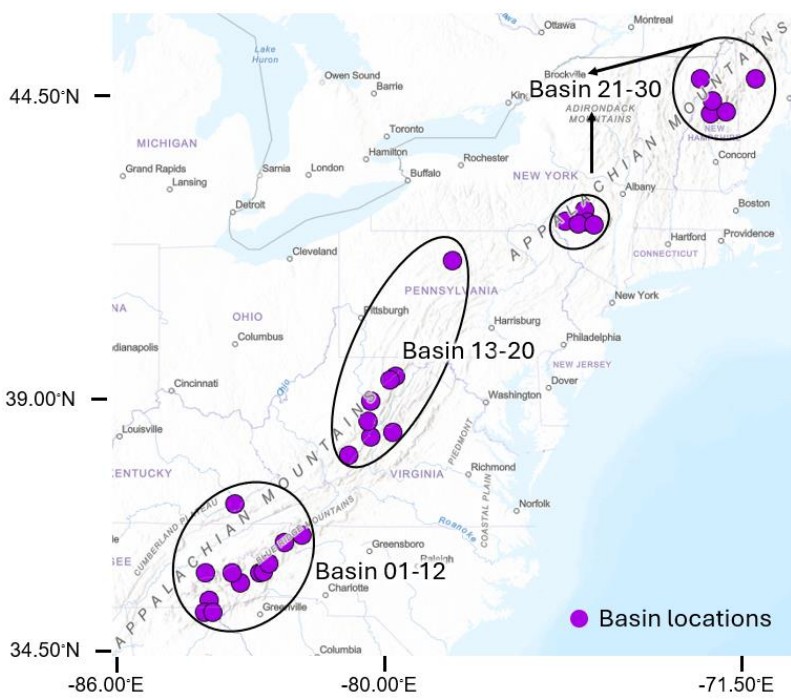

**Figure 4** – Map of the Continental United States (CONUS) and headwater basins studied in this work. Basin information is available in Table 3. Sub-regions are delineated as the following for discussion purposes only: Northern, Central and Southern Appalachian Mountains (NAM, Basin 21-30; CAM, Basin 13-20; SAM, Basin 01-11). This figure is adapted from Liao and Barros (2025b).



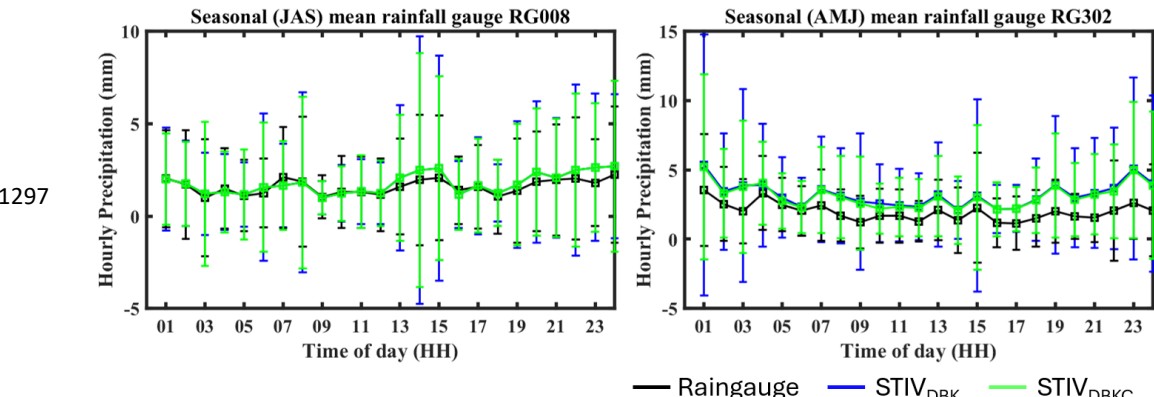

**Figure 5** – Examples of raingauge measurements showing the diurnal cycle of different seasons at different locations: Left panel – raingauge RG008 located in the eastern ridges for the Summer (JAS: July-August-September) season. Right panel – raingauge RG302 located in the western ridges for the Spring (AMJ; April-May-June) season. Rain gauge measurements (blue); StageIV$_{DBK}$ (black); StageIV$_{DBKC}$ (green). This figure is from Liao and Barros (2019).



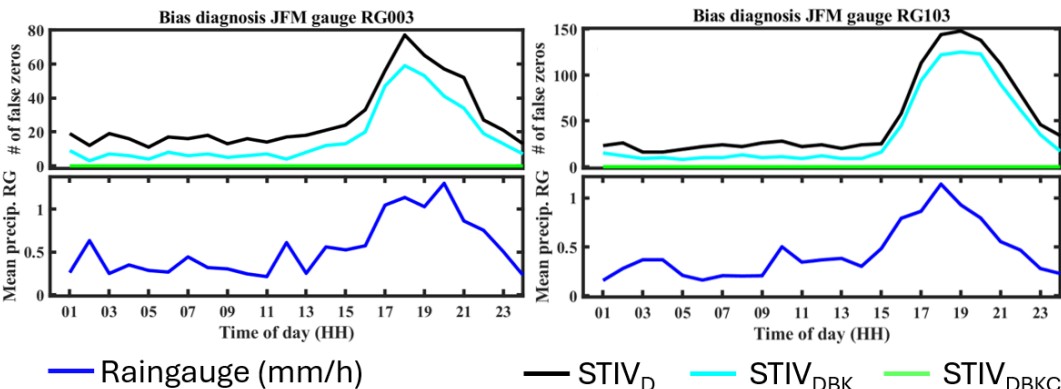

**1304**

**1305** **Figure 6** –Top row – The diurnal cycle of missing precipitation at RG003 (Eastern ridges) and
**1306** RG103 (Inner regions) for January-February-March (JFM) using various products. Bottom row-
**1307** corresponding rain gauge climatology (blue). StageIV$_D$ (black); StageIV$_{DBK}$ (cyan); StageIV$_{DBKC}$
**1308** (green). This figure is from Liao and Barros (2019).

**1309**



1310

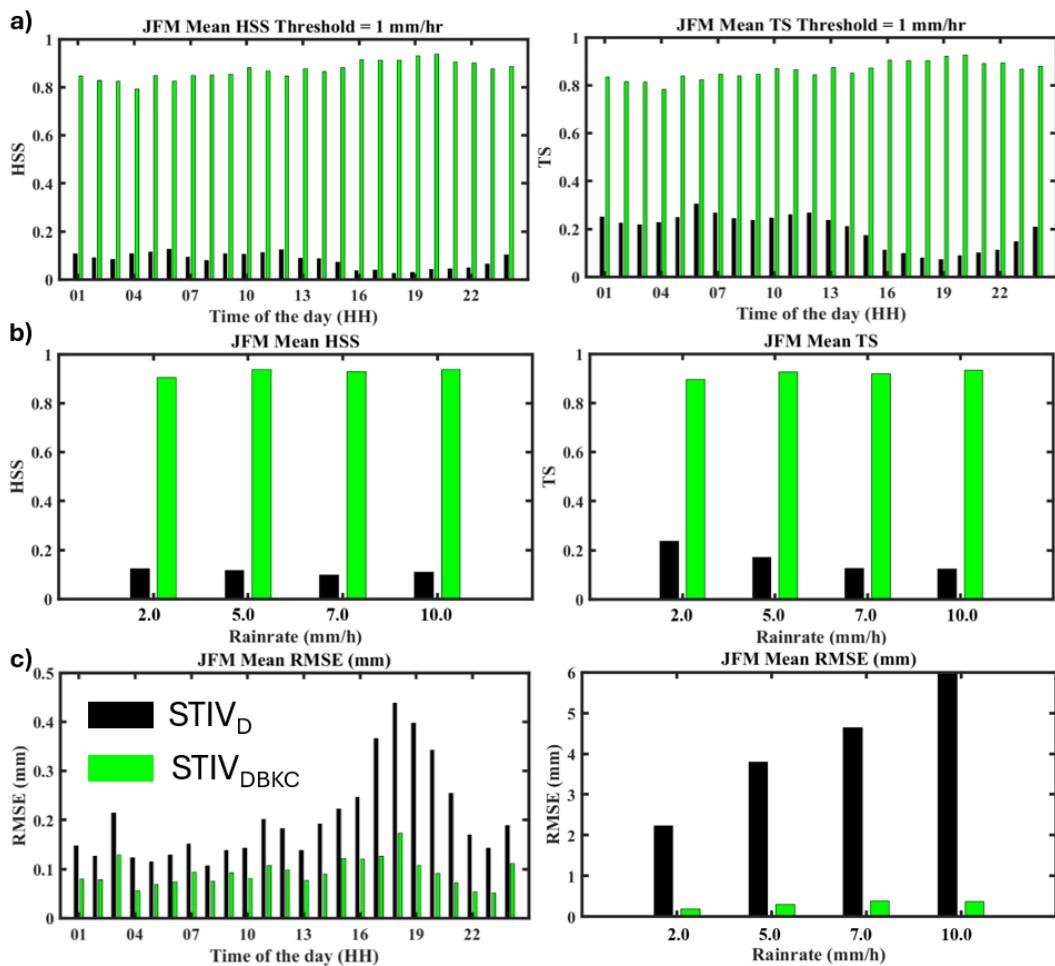

1311

**Figure 7** – Statistical evaluation summary for winter precipitation (JFM, January, February, and March): a) Diurnal cycle of mean HSS and TS statistics including all rain gauges calculated using all data from 2008 to 2017: $STIV_D$ (black) and $STIV_{DBKC}$ (green); b) HSS and TS statistics calculated using different rain rate thresholds over the same 10-year period; c) Diurnal cycle of rain rate RMSE at seasonal-scale, and its dependence on observed rainfall rate. This figure is from Liao and Barros (2019).

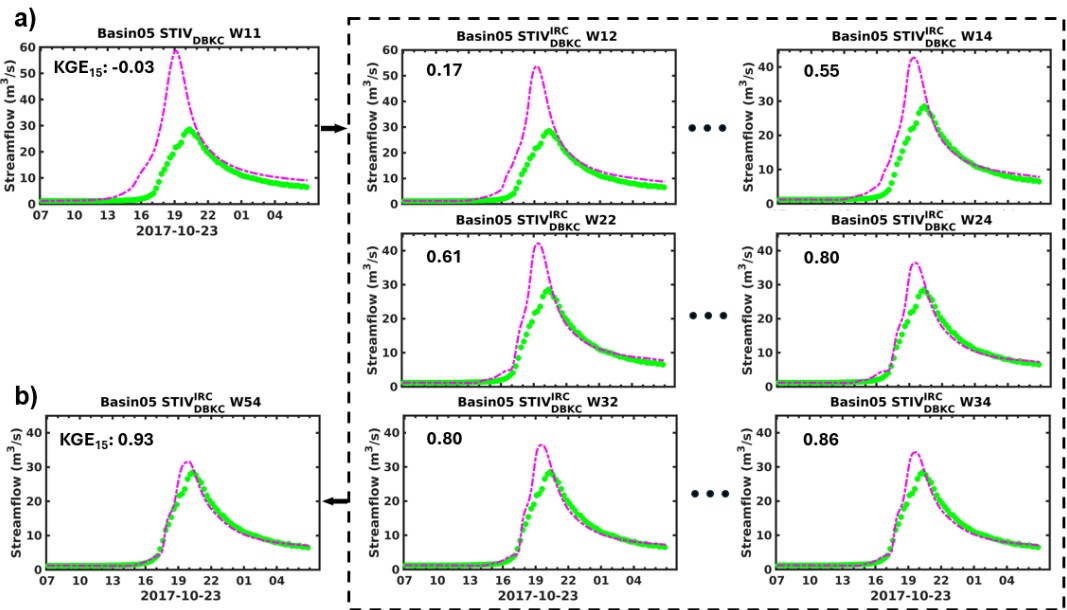

1318

**Figure 8** – The IRC-ICC performance in Basin05 as an example for the 2017-10-23 event (Basin05: Cataloochee Creek Basin, NC). This event is part of the 2017 Hurricane Nate. This figure contains **a)** hydrological responses when precipitation forcing is the $STIV_{DBKC}$. The dashed rectangular plot consisting of intermediate results including each iteration from the IRC-ICC framework (Figure 3). **b)** the hydrological equilibrium of the IRC-ICC after 5 iterations. This figure is adapted from Liao and Barros (2025b).

1325

Open Access · Earth System Science Data Discussions

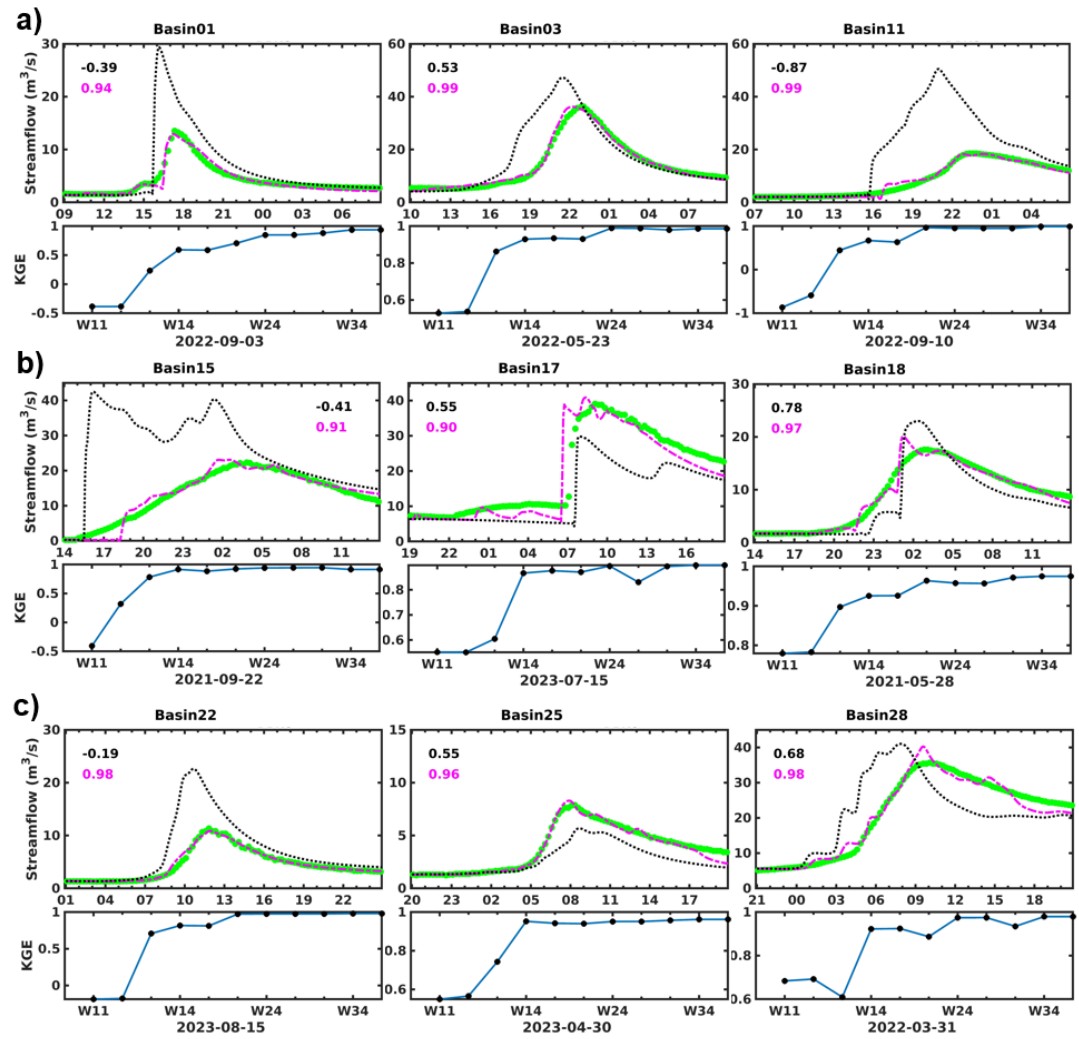

**Figure 9** – The IRC-ICC performance for different subregions, include **a)** 3 events from the Southern Appalachians; **b)** 3 events from the Central Appalachians; and **c)** 3 events from the Northern Appalachians. The IRC-ICC KGE evolution plots from iterations are included below the hydrographs. The black and pink line are from the original $STIV_D$ and the IRC-ICC equilibrium state ($STIV_D^{IRC*}$), respectively, and KGE values are displayed as colored numbers in the top left corners. This figure is adapted from Liao and Barros (2025b).

1333

1334





**Figure 10** – Event total precipitation maps for three cold season events (**a**) and three warm season events (**b**). Each column represents one event, and each row represents one precipitation product: $\text{STIV}_{\text{DBKC}}$, $\text{STIV}_{\text{DBKC}}^{\text{IRC}*}$ from IRC-only framework, and $\text{STIV}_{\text{DBKC}}^{\text{IRC}*}$ from the coupled IRC-ICC framework. This figure is adapted from Liao and Barros (2025b).


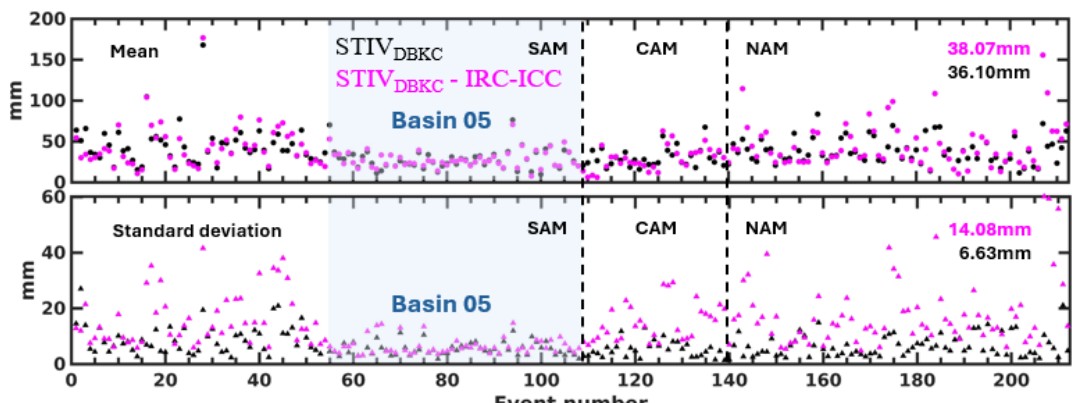


**Figure 11** – Summary charts of precipitation statistics for all event-total precipitation maps. Basin
precipitation average and standard deviation for each event are represented by circles and triangles
in the top and bottom panel, respectively. Each panel consists of 3 sub-regions by vertical black
lines: the Southern Appalachian Mountains, Central Appalachian Mountains, and Northern
Appalachian Mountains. The list of events in Basin 05 (with event number ranging from 55 to
108) in the SAM is highlighted by a blue rectangle for further discussion in the text. The average
values of all events for both the mean and the standard deviation are calculated and shown in the
top right corner. Black color and pink color represent pre and post IRC-ICC QPE statistics,
respectively.




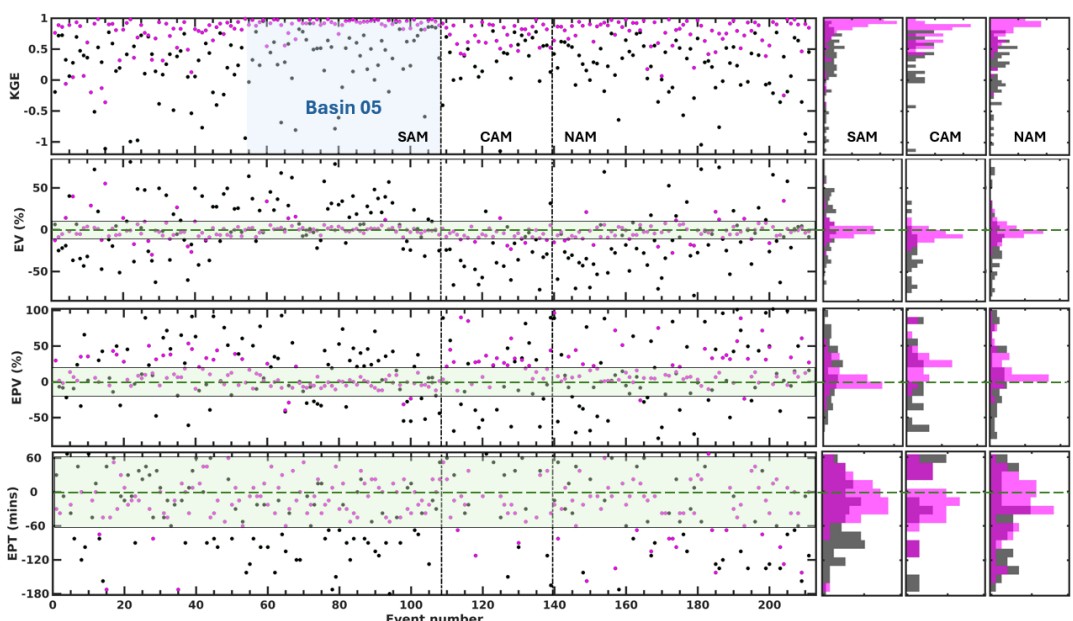


**Figure 12** – Summary of hydrologic skills. Green dashed lines and associated uncertainty envelopes are only for visual illustration. Hydrologic statistics are explained in Table 2. Pink and black scatters (each scatter represent one event) represent IRC-ICC, and baseline outputs, respectively. Each horizontal panel is split into 3 subsections by vertical black lines representing the 3 subregions. Histograms graphs on the right hand side are provided for a summary view. This figure is adapted from Liao and Barros (2025b).







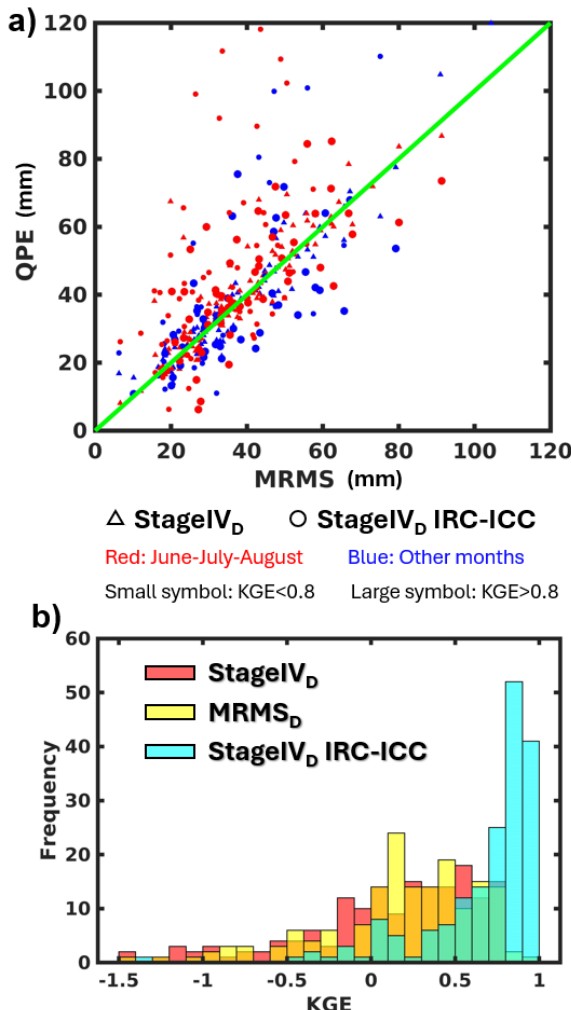


**Figure 13 – a)** Event total QPE plots for various QPE datasets conditional on seasons and KGE values; **b)** KGE distributions across events using different QPE datasets. This figure is adapted from Liao and Barros (2025b).