# Peer review of "StageIV-IRC: A High-resolution Dataset of Extreme Orographic Quantitative Precipitation Estimates (QPE) Constrained to Water Budget Closure for Historical Floods in the Appalachian Mountains"

_Earth System Science Data, 2025_

## Author Comment (AC1)

We thank the Reviewer for the comments.  Our detailed replies are in blue below.

**RC1: 'Comment on essd-2025-554', Anonymous Referee #1, 02 Oct 2025**

The author developed a High-resolution Dataset of Extreme Orographic QPE by closing the water budget using stream gauge measurements. This is a novel method and will be of great value if further validated. Therefore, I recommend a major revision, as some clarification is needed, and more dataset evaluation may be beneficial.

Major comments:

1. I would recommend that the authors mention ICC as well in the abstract, as it is also one step in the precipitation data generation.

Thank you. The abstract is now revised with information regarding the ICC.

2. I recommend that the author provide a brief code to show how to read the data. The current format and structure of the data are unclear. It will be helpful for readers to try the data.
A simple Matlab script to read the QPE data is provided in the same Zenodo repository as a new version.

3. Are the ICC and IRC corrections implemented simultaneously in windows 2 and 5? Intuitively, overestimated rainfall values can compensate for an underestimated initial soil moisture condition. I am curious whether this compensation causes some difficulties in determining precipitation.

ICC and IRC are simultaneously implemented in windows 2 and 5. It is assumed that rainfall uncertainty dominates uncertainties in windows 3 and 4 (flood high flows). Therefore, initial soil moisture is assumed to dominate hydrologic uncertainties in windows 2 and 5. To reflect the priority of ICC in these two windows (2 and 5), at each time step in these two windows, soil moisture correction is carried out first, and then the remaining hydrologic residuals are attributed to rainfall uncertainty as documented in Liao and Barros in detail (2025b, JoH Reg. Stud.).

4. In the inverse correction process, there are likely more unknowns (precipitation at each pixel) than the knowns (observed discharge). Is it possible to obtain two different precipitation fields that can generate very similar discharge? How can you guarantee that you can get the "optimal" precipitation fields compared to other possible realizations? Is it reasonable to obtain an ensemble precipitation dataset to account for this variability?

Thank you for the insightful comments. The inverse correction process itself is a deterministic process for a given original precipitation in this case StageIV. However, the fractal downscaling method used to bring precipitation from the original StageIV resolution (4km) to the resolution of the precipitation product in this work, is statistical in nature and it is required that an ensemble of precipitation fields be generated (Bindlish and Barros, 2000; Tao and Barros, 2010; Nogueira and Barros, 2015). In the current implementation of the IRC-ICC, we then take the ensemble mean and that is the precipitation forcing that is being corrected. Therefore, instead of using the ensemble mean, it is possible to use the full ensemble. The challenge is that for each ensemble member, it would need to determine the distribution of travel times and associated trajectories for Lagrangian backtracking of the error as per the IRC. This is computationally very demanding and by and large prohibitive. Because of the very small area of headwater basins, the differences among ensemble members are very small (see also references above) and thus the variance among ensemble members is very small as shown also in Tao and Barros (2010). As we apply the IRC-ICC to larger river basins, this question should be considered more carefully. A statement to this effect was added to the manuscript.

5. Why did the authors select Stage IV as the primary precipitation source? In the first step, the authors downscale the precipitation field from 4km to 1km. Other available precipitation datasets, such as MRMS and AORC, provide precipitation estimates at a 1km resolution. If the authors use these 1km datasets, the downscale step can be removed.

The authors selected StageIV as the basis because StageIV is widely considered the best long-term gridded precipitation dataset for the U.S., as we have flood events in this study dated back to 2008 and thus the length of the record is critical. In addition, products like MRMS suffer from the same spatial biases in the mountains as does StageIV. The authors used MRMS as a 'reference' dataset for comparison purposes, as shown in Figure 13. The AORC is a reanalysis dataset that contains many data sources, which makes it hard to extract useful information on rainfall estimation errors.

6. L201-204, what does "self-similar statistics" mean? In L213, what does "the same rainfall statistics" mean here? I am curious which type of rainfall statistics is preserved in the downscaling process.
The self-similar statistics specifically refer to the power-law-like behavior of rainfall fields in the Fourier domain, which is a widely observed behavior (Nogueira and Barros, 2015, Eq. 5). This scaling behavior enables downscaling across scales, that is to generate stochastic rainfall fields that preserve the same scaling behaviors. In other words, this scaling

behavior (i.e., described by the slope of the power spectrum, the beta value in Eq. 5) is the key rainfall statistic that is preserved from coarser scales to finer scales.

7. What is the size of the rainfall field in Ordinary Kriging? Is it a basin-based correction? Ordinary Kriging has the assumption of geostationary, which may not perform optimally when applied to a large complex region.

The Kriging approach was done over a 1-degree x 1-degree box that contains the raingauge network in the Southern Appalachians. Since this area is relatively small, we expect the impact of rainfall heterogeneity to be acceptable when using Kriging.

8. L505-L508, the authors mentioned that "The climatologically corrected STIV_DBKC fields have a significantly accurate diurnal cycle compared to only event-scale bias-corrected STIV_DBK." But in Figure 5, I did not see many differences between the blue and green lines. And should not the "STIV_DBK" here be "STIV_DB"?

Thank you for pointing out this mistake. The legend was wrong. The blue is the rain gauge and the green is STIV_DBKC, and the black is the STIV_DBK. It is now fixed. Here in Figure 5, we exclude the potential impact of Kriging and only focus on the impact of climatology corrections, and that's why we are comparing STIV_DBK against STIV_DBKC. This figure shows that STIV_DBK consistently underestimates rainfall (by 0.5mm-1mm/hr) climatologically for a 10-year study period, and Kriging alone could not resolve this underestimation.

9. L610, the authors mentioned that "IRC-ICC" is the recommended dataset. In Section 5, the author provides the citation for "IRC". Why don't the authors publish IRC-ICC?

Thank you. The dataset we published is IRC-ICC, and we added a couple of sentences in the abstract to include the step 'ICC' following Major comment 1. In general, the improvement in hydrographs from IRC-only to IRC-ICC is relatively small compared to from original rainfall to IRC-only because rainfall uncertainties dominate over initial soil moisture uncertainties for these significant flood events (Figure 5 in Liao and Barros, 2025b), which also means smaller rainfall changes from IRC-only rainfall to IRC-ICC rainfall. In summary, ICC is only one component of the IRC framework aimed at reducing the impact of initial soil moisture uncertainty impacting mostly the time to rise. Other components in the general IRC framework, such as improved routing schemes and land

surface parameterizations, are also published (Liao and Barros, 2025a, *WRR*). This is also why we name the dataset StageIV-IRC.

10. I recommend that the authors provide the results of STIV_IRC_ICC in Figures 5, 6, and 7. I understand that the lack of rainfall ground truth makes the evaluation of precipitation data a little bit hard. The better discharge estimates from your methods cannot reflect the absolute accuracy of precipitation data, as the discharge is your objective function. I would recommend more evaluation of the precipitation data itself. Alternatively, you can use STIV_IRC_ICC to drive another hydrologic model to evaluate whether you can also have a better discharge prediction than Stage IV. Model calibration can also be implemented, as hydrologists usually do so with a precipitation dataset.

Figures 5, 6, and 7 are climatology results, but the IRC-ICC dataset is implemented at flood-event scales. In principle, IRC-ICC can be applied to the entire record of radar rainfall estimation, but it is not computationally feasible for us. We do have some verification against other precipitation data sources using the IRC-only approach (e.g., Figure 9 in Liao and Barros, 2022, *RSE*) and the IRC-ICC approach (Liao and Barros, 2025b, *JoH Reg. Stud.*). It is worth noting that, apart from the scarcity of raingauges in this region, almost all raingauges are located at the ridges (i.e., along the borders of the basins, see the zoom-in figure below for Basin05 or CCB as an example), which makes it challenging for a fair rainfall comparison for the IRC framework because IRC is based on travel time theory and rainfall at the basin borders can take days to reach the basin outlet even for flood-producing events. This is beyond the current simulation and computation capabilities using the IRC (i.e., 24 hours). Further, rainfall comparison at point scales (e.g., raingauges) requires much finer spatial resolution of the IRC-ICC, as rainfall heterogeneity is significant in this region.

[Figure]

The reason that calibration is not involved in our study is that parameter calibration often compensates for precipitation uncertainties. Lots of studies show poor hydrological

performance of a calibrated model when evaluated outside the calibration period, and flood peaks are often poorly captured. We propose this is due to the lack of a proper reduction in precipitation uncertainty, which is important for high-flow floods.

In terms of using IRC-ICC products to drive another model, we will have to leave this to the readers, which is why we are releasing the data. Different hydrological models have different strengths. This dataset is specifically developed for extreme flood events in complex terrain. Therefore, many aspects of a hydrological model can impact the performance. For example, if a hydrological model has a less advanced routing module specifically built for steep-slope terrain, the results will reflect that.  We would recommend that our proposed XY routing scheme (Liao and Barros, 2025a), which is trivial to implement,  be adopted by all models in steep terrain.  The DCHM used in this study has developed and evolved starting the mid-90s, with a particular focus on complex terrain.

Minor comments:

1. I recommend that the authors clarify the terminology usage. In Figure 2, the event scale bias correction is noted as STIV_BD. But in some places of the figure and the article, STIV_DBK is used.

STIV_DBK was also created (not shown in Figure 2) for the purpose of illustrating the importance of climatological adjustments by comparing STIV_DBK against STIV_DBKC. Otherwise, it is not fair to compare STIV_DB against STIV_DBKC for climatological evaluations.

2. L690.  The resolution of StageIV_D is "1km, hourly" in Figure 1, but you mention " the same resolution as StageIV_D datasets (250m, 5min)".

In lines 541-542, we explained that for hydrological simulation, StageIV_D is further downscaled to 250m resolution. This is to capture the fast response in flash floods. Following this comment, we recognized that this is confusing and added clarification at Line 690.

3. Provide the legend in Figure A3, Figure 8,9, 11, 12

4. Provide the unit in Figure 10

5. Provide the y-axis in Figure 11

Thank you. The figures are revised according to the three comments above.

---

## Author Comment (AC2)

We thank the Reviewer for the comments. Our detailed replies are in blue below.

**RC2: 'Comment on essd-2025-554', Anonymous Referee #2, 21 Nov 2025**

Excellent work! My only concern is regarding the Inverse Correction process. What is the solution process given that there is more precipitation data than observed discharge? is there an averaging process of the rainfall in the basin ? I think this is a similar question to comment 4 by reviewer 1. Thanks !

Thank you for your question on the inverse rainfall correction. It seems that there is more precipitation information than discharge information. However, water travel time distributions are integrated in space and time, and the final results are the basin discharge. Therefore, if basin discharge went through the de-convolution process, the water budget residual errors would be revealed as precipitation uncertainties. In the current version of the inverse rainfall correction, water budget residual errors are distributed in a deterministic way, and it is based on precipitation magnitude at the location mapped by the Lagrangian backward tracking. Therefore, there is no averaging process in the approach, but every pixel gets different corrections based on the rainfall intensity at their location.